# Cirrus cloud retrieval with MSG/SEVIRI using artificial neural networks

Johan Strandgren[1], Luca Bugliaro[1], Frank Sehnke[2], and Leon Schröder[2]

[1]Deutsches Zentrum für Luft- und Raumfahrt, Institut für Physik der Atmosphäre, Oberpfaffenhofen, Germany
[2]Zentrum für Sonnenenergie- und Wasserstoff-Forschung Baden Württemberg, Systemanalyse, Stuttgart, Germany
*Correspondence to:* Johan Strandgren (johan.strandgren@gmail.com)

**Abstract.** Cirrus clouds play an important role in climate as they tend to warm the Earth-Atmosphere system. Nevertheless their physical properties remain one of the largest sources of uncertainty in atmospheric research. To better understand the physical processes of cirrus clouds and their climate impact, enhanced satellite observations are necessary. In this paper we present a new algorithm, CiPS (Cirrus Properties from SEVIRI), that detects cirrus clouds and retrieves the corresponding cloud top height, ice optical thickness and ice water path using the SEVIRI imager aboard the geostationary Meteosat Second Generation satellites. CiPS utilises a set of artificial neural networks trained with SEVIRI thermal observations, CALIOP backscatter products, the ECMWF surface temperature and auxiliary data.

CiPS detects 71 % and 95 % of all cirrus clouds with an optical thickness of 0.1 and 1.0 respectively, that are retrieved by CALIOP. Among the cirrus free pixels, CiPS classifies 96 % correctly. With respect to CALIOP, the cloud top height retrieved by CiPS has a mean absolute percentage error of 10 % or less for cirrus clouds with a top height greater than 8 km. For the ice optical thickness, CiPS has a mean absolute percentage error of 50 % or less for cirrus clouds with an optical thickness between 0.35 and 1.8, and of 100 % or less for cirrus clouds with an optical thickness down to 0.07, with respect to the optical thickness retrieved by CALIOP. The ice water path retrieved by CiPS shows a similar performance, with mean absolute percentage errors of 100 % or less for cirrus clouds with an ice water path down to $1.7 \, \mathrm{g \, m^{-2}}$. Since the training reference data from CALIOP only include ice water path and optical thickness for comparably thin clouds, CiPS does also retrieve an opacity flag, which tells whether a retrieved cirrus is likely to be too thick for CiPS to accurately derive the ice water path and optical thickness.

By retrieving CALIOP like cirrus properties with the large spatial coverage and high temporal resolution of SEVIRI during both day and night, CiPS is a powerful tool for analysing the temporal evolution of cirrus clouds including their optical and physical properties. To demonstrate this, the life cycle of a thin cirrus cloud is analysed.

## 1 Introduction

High level clouds cover 27-37 % of the Earth's surface (Stubenrauch et al., 2013, the exact figure depending on the satellite instrument and its sensitivity to thin and sub-visual cirrus) and do consequently play an important role in the climate system by reflecting the incoming solar radiation and absorbing the outgoing thermal radiation. In this study we focus on cirrus clouds,

defined as all clouds that consist of ice crystals. In general, the net cirrus radiative forcing is strongly depending on the position and thickness of the cloud as well as the microphysical properties like ice crystal shape, size distribution and ice water content (e.g. Fu and Liou, 1993; Zhang et al., 1999; Liou, 2002; Wendisch et al., 2007). Because of the tenuous nature of cirrus clouds the reflection of solar radiation can be outweighted by the thermal effect (Meerkötter et al., 1999) leading to a positive net

radiative forcing, as is the case for thin cirrus (Jensen et al., 1994; Chen et al., 2000). Despite the constant progress in cirrus research and the continuous development of more advanced instruments and retrieval algorithms, the understanding of the physical processes that govern the cirrus life cycle as well as the temporal evolution of their physical and optical properties is still limited, as is their representation in weather and climate models (Waliser et al., 2009; Eliasson et al., 2011).

To capture the temporal evolution throughout the cirrus life cycle as well as the diurnal cycles of cirrus coverage and

properties like cloud top height (CTH), ice optical thickness (IOT) and ice water path (IWP), it is essential to accurately and consistently detect and monitor cirrus during both day and night. To this end, imagers like SEVIRI (Spinning Enhanced Visible and Infrared Imager, Schmetz et al., 2002) aboard the geostationary Meteosat Second Generation (MSG) satellites are the instruments of choice since they combine a large field of view with a high temporal resolution.

Cirrus clouds can be detected from space-borne imagers (e.g. Saunders and Kriebel, 1988; Derrien et al., 1993; Ackerman

et al., 1998; Kriebel et al., 2003; Derrien and LeGleau, 2005; Krebs et al., 2007) by applying spectral tests on brightness temperatures and temperature differences (e.g. Inoue, 1985; Ackerman et al., 1990). Krebs et al. (2007) extend the multi-spectral threshold test approach by introducing morphological tests that take into account the shape of high level clouds in thermal water vapour observations. Near infrared water vapour absorption channels can also be used to detect cirrus clouds (Gao et al., 2002). Passive imagers do however have a limited sensitivity to thin cirrus clouds and algorithms utilising spectral

and morphological threshold tests tend to miss a large fraction of those thin cirrus (e.g. Ackerman et al., 2008; Holz et al., 2008; Stubenrauch et al., 2010) and thus introduce a bias into the climate impact of cirrus clouds. Another well known problem related to cloud detection from passive imagers is the difficulty to distinguish between cirrus clouds and cold surfaces in the polar regions (e.g. Holz et al., 2008).

The CTH is an important variable as it determines the outgoing longwave radiation. It can be retrieved from passive satellite

imagers during both day and night using e.g. radiance ratioing (also referred to as $CO_2$ absorption, $CO_2$ slicing and split window technique) (Smith et al., 1970; Smith and Platt, 1978; Menzel et al., 1983; Eyre and Menzel, 1989; Zhang and Menzel, 2002; Menzel et al., 2008), radiance fitting (e.g. Szejwach, 1982; Nieman et al., 1993; Schmetz et al., 1993), and optimal estimation (e.g. Heidinger and Pavolonis, 2009; Sayer et al., 2011; Watts et al., 2011). An inter-comparison of different techniques currently used for SEVIRI is presented in Hamann et al. (2014).

Nakajima and King (1990) introduced a commonly applied approach for the retrieval of optical thickness and effective particle radius of clouds from reflected solar radiation in two spectral channels (e.g., Platnick et al., 2003; Bugliaro et al., 2011; Stengel et al., 2014) for both ice clouds and liquid water clouds. From the optical thickness and effective radius the liquid and ice water paths (LWP and IWP respectively) can be estimated for liquid and icy pixels respectively. The solar dependence does however limit this approach to daytime and the retrieval becomes ambiguous for optically thin clouds (Nakajima and King,

1990). The same properties can be retrieved for optically thin cirrus clouds during night as well using only thermal observations

(e.g. Prabhakara et al., 1988; Ackerman et al., 1990; Yue and Liou, 2009; Minnis et al., 2011; Heidinger et al., 2015; Wang et al., 2016), but with a limited accuracy due to the low sensitivity to large ice crystal sizes and large optical thicknesses.

The limited amount of vertical information and sensitivity to thin cirrus clouds is a recurrent drawback of passive imagers. The spaceborne lidar CALIOP (Cloud-Aerosol Lidar with Orthogonal Polarization, Winker et al., 2009), measures profiles of attenuated backscatter with a vertical resolution of up to 30 m and is currently the most accurate source for the detection of cirrus clouds and the retrieval of their top height and optical thickness from space. CALIOP is an active sensor and can consequently operate during both day and night but the small spatial scale and the repeat cycle of approx. 16 days make it inadequate for studying the temporal evolution of cirrus clouds.

As an attempt to combine the advantages from a polar orbiting lidar and a geostationary imager, Kox et al. (2014) present an approach for the detection and retrieval of optical thickness and top height of cirrus clouds from SEVIRI. Their algorithm COCS (Cirrus Optical properties from CALIOP and SEVIRI) utilises an artificial neural network (ANN) trained with coincident CALIOP backscatter and SEVIRI thermal observations in order to estimate CALIOP-like cirrus properties from SEVIRI. During the training procedure the ANN learns to generalise, such that it can estimate a desired output vector for a set of previously unseen input data. This together with the low computational costs makes neural networks an interesting alternative to more commonly used physically based methods. Minnis et al. (2016) present a similar approach to estimate the optical thickness of opaque ice clouds at night using an ANN trained with coincident CloudSat/CPR (Cloud Profiling Radar) measurements and Aqua/MODIS (Moderate Resolution Imaging Spectroradiometer) infrared radiances. Holl et al. (2014) use combined CALIPSO/CALIOP and CloudSat/CPR retrievals for the retrieval of the IWP from AVHRR (Advanced Very High Resolution Radiometer) and MHS (Microwave Humidity Sounder) on the NOAA and MetOp satellites using neural networks. Cerdena et al. (2006, 2007) use neural networks trained with simulated radiances for the retrieval of optical thickness, effective radius and temperature of liquid water clouds (day and night) and cirrus clouds (only day) from NOAA/AVHRR. Taravat et al. (2015) use neural networks for the daytime cloud detection from SEVIRI.

In this paper we present CiPS (Cirrus Properties from SEVIRI), a new algorithm for cirrus remote sensing with SEVIRI that exploits the basic idea of COCS: retrieving cirrus properties using ANNs trained with CALIOP and SEVIRI data. However, CiPS clearly differs from COCS in the implementation of this idea and the achieved performance. For a more accurate cirrus detection and determination of CTH and IOT, CiPS utilises a different set of input parameters including numerical weather model data and information about nearby pixels. In addition, CiPS classifies each pixel as either cirrus free, transparent cirrus or opaque cirrus by means of dedicated classification ANNs. As CALIOP gets saturated for thicker clouds, the opacity information is an important additional piece of information in order to better characterise the cirrus and the reliability of the ANN results that was absent in COCS. Furthermore, CiPS is trained to retrieve the IWP, resulting in a total of three climate relevant cirrus cloud properties that can be estimated during both day and night for the full SEVIRI field of view every 15 minutes. In particular, the IWP retrieved by CiPS allows for a direct comparison with climate, weather and large eddy simulation models. CiPS targets thin cirrus clouds, as those clouds are most difficult to retrieve using thermal satellite observations from geostationary orbits. The more thin cirrus clouds that can be detected and accurately retrieved, the smaller the bias of the derived radiative forcing and climate impact of cirrus clouds will be. Thus CiPS helps to fill this gap of observations in cloud remote sensing.

The remainder of this paper is divided into five parts. In Sect. 2 the instruments, data and tools used for this study are introduced and described. The new algorithm, CiPS, is described in detail in Sect. 3. Section 4 shows the performance of CiPS for a SEVIRI scene over parts of Europe together with a detailed validation of all quantities using CALIOP as reference. To illustrate the capability and performance of CiPS, a life cycle analysis of a thin cirrus cloud using CiPS is presented in Sect. 5. Finally the performance of CiPS is shortly summarised and discussed in the concluding section. A list of abbreviations is available in Appendix A.

## 2   Instruments and tools

### 2.1   SEVIRI

SEVIRI is a passive imager operating aboard the geostationary Meteosat Second Generation (MSG) satellites operational since 2004. SEVIRI is positioned at $0°$ E (operational service) and has an excellent view of the Earth from its remote location, with a spatial coverage from approx. $80°$ W to $80°$ E and $80°$ S to $80°$ N. SEVIRI has a sampling distance of 3 km at nadir (1 km for the broadband visible channel) and a temporal resolution of 15 min. Limiting the spatial coverage to the upper part of the SEVIRI disc (north of approx. $15°$ N), the temporal resolution can be increased to 5 min using the rapid scanning service. SEVIRI measures the up-welling radiation in twelve wavelength intervals (Schmetz et al., 2002), from which the radiances, reflectances and equivalent black body brightness temperatures can be derived.

### 2.2   CALIOP

CALIOP was launched as the main instrument aboard the CALIPSO (Cloud-Aerosol Lidar and Infrared Pathfinder Satellite Observations) satellite in 2006. CALIPSO is flying in a sun-synchronous orbit as part of the A-Train (Stephens et al., 2002). CALIOP is an elastic backscatter lidar operating at 2 wavelengths, 532 and 1064 nm. By emitting approx. 20 laser pulses per second, a ∼70 m footprint is produced every 335 m on the Earth's surface, resulting in curtains of attenuated backscatter profiles along the CALIPSO track (Winker et al., 2009). A long set of algorithms are applied to the backscatter profiles in order to detect cloud and aerosol layers (Vaughan et al., 2009), differentiate between the two (Liu et al., 2009), determine the cloud phase (Hu et al., 2009) and finally derive profiles of volume extinction coefficients (Young and Vaughan, 2009). For the cloudy regions where the cloud phase is determined to be ice, the ice water content (IWC) is calculated from the retrieved extinction coefficients using a parametrisation derived by Heymsfield et al. (2005) based on extensive in situ measurements. The layer IOT (ice optical thickness) and IWP is obtained by integrating the vertical profiles of extinction coefficients and IWC.

### 2.3   Artificial neural networks

An artificial neural network consists of a number of neurons that exchange information with each other, in a similar manner as biological nerve cells transmit information via synapses in the human brain. By assigning each neuron-neuron connection a numeric tunable weight, the ANN has the ability of learning patterns and approximating functions. The goal of an ANN is to

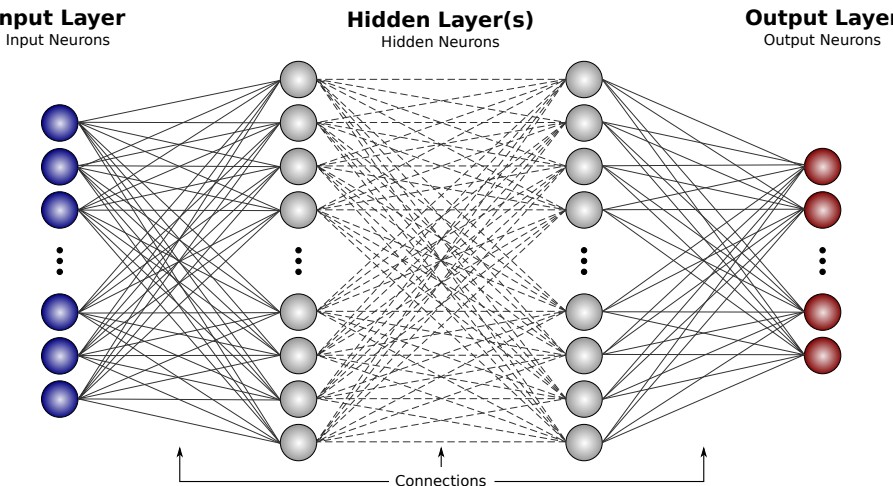

**Figure 1.** Generic structure of a multi layer perceptron (MLP), a form of a feed-forward artificial neural network used in this study.

derive a vector of unknown output variables given a vector of known input data. This tool is applied in Sect. 2.5 and 3 to the remote sensing of cirrus clouds and is thus introduced in the following.

### 2.3.1 Multilayer perceptron

In this study a multilayer perceptron (MLP), a feed-forward artificial neural network, is used. A MLP consists of three major units; 1) the input layer, 2) the output layer and 3) the hidden layer(s). The input layer holds as many neurons as input variables and the output layer as many neurons as desired output variables. The hidden layer(s) hold an arbitrary number of additional neurons distributed over an arbitrary number of hidden layers. All connections between the neurons within the MLP are in the forward direction (input layer → hidden layer(s) → output layer). Connections backward or within a layer are forbidden (Rumelhart et al., 1986). The value of a neuron is calculated by processing the output from the preceding neurons connected to that neuron and the corresponding weights through an *activation function*. These non-linear functions allow the ANN to solve complex problems with a limited number of neurons. A generic structure of a MLP is illustrated in Fig. 1. In addition to the input and hidden neurons, a constant bias neuron is commonly added to the input and hidden layers in order to give the MLP more flexibility during the training.

When the MLP is given a vector of input data it uses the connection weights and possible biases to estimate the vector of output data. Thus, it is crucial that the weights and bias neurons are assigned correct values.

### 2.3.2 Learning through backpropagation

The weights are tuned by training the MLP, which is done with a teacher/trainer approach, more known as supervised training. A commonly used training algorithm is the backpropagation algorithm. The most essential steps in the backpropagation algorithm are explained below, but for the curious reader the algorithm as a whole is well explained in Rumelhart et al. (1986).

Using backpropagation the network is fed with a set of training examples where the vector of input variables as well as the vector of expected output variables are known. From the training input data the MLP estimates its own output data using the current weights. From the vector of *estimated* output and the corresponding vector of *expected* output the total error $E$ (squared difference) is calculated. The error is then propagated backwards through the MLP and used to update each weight using gradient descent in such a way that the total error is minimised. Each weight is updated using the following equation:

$$w_{ij}^* = w_{ij} - \eta \frac{\delta E}{\delta w_{ij}} = w_{ij} - \Delta w_{ij} \, . \tag{1}$$

where $w_{ij}$ and $w_{ij}^*$ are the old and new values for a weight connecting the two neurons $i$ and $j$. $\frac{\delta E}{\delta w_{ij}}$ describes how much a change in $w_{ij}$ affects the total error $E$. To adjust how aggressive the weight updates should be, a *learning rate $\eta$* is multiplied with $\frac{\delta E}{\delta w_{ij}}$ before the weight update. A larger learning rate means larger changes in the weights and thus a faster training. This can however lead to an oscillation of the total error around a minimum solution. With a small learning rate the total error will not oscillate around a minimum solution, but the training is slower and the risk of getting stuck in local minima is higher. By introducing a *momentum* term $\alpha$, possible oscillations in the iterative search for the minimum error are attenuated, and this allows for a larger learning rate. The momentum makes use of the previous update of the corresponding weight in order to get a weighted sum of the current and previous error gradients. The momentum term is added to the second term on the right hand side of Eq. (1) such that

$$\Delta w_{ij}^k = \eta \frac{\delta E}{\delta w_{ij}} + \alpha \Delta w_{ij}^{k-1} \, , \tag{2}$$

where $k$ represents the $k^{th}$ update of the weight $w_{ij}$, meaning that $\Delta w_{ij}^{k-1}$ is the previous update of weight $w_{ij}$ (Rumelhart et al., 1986).

To find the minimum total error between the estimated and expected output vectors for a complex problem and tune the weights accordingly, a large training dataset is required. Training an MLP is an iterative process, where each training example is presented to the ANN multiple times until a satisfying result has been achieved. With common ANN terminology the training completes one *iteration* every time the weights are updated and one *epoch* when all training examples contained in the training dataset have been presented to the ANN. The amount of iterations required for one epoch does therefore depend on the amount of training examples the ANN is given for every update of the weights i.e. the *batch size*. With stochastic gradient descent (sometimes referred to as momentum stochastic gradient descent, when the momentum term is used) the weights are updated for each training example (batch size $= 1$), whereas for full batch gradient descent the weights are updated using all training examples at once (batch size $= N$, where $N$ is the total number of training examples). Stochastic gradient descent leads to a noisy error gradient whereas the full batch gradient descent requires more computational power to converge. With mini-batch gradient descent an intermediate number of training examples is used for each weight update ($1 <$ batch size $< N$).

While in recent years very potent new learning methods that are based on backpropagation were developed, stochastic gradient descent is still the most used method due to its simplicity and robustness (Schmidhuber, 2015).

**Table 1.** Contingency table for the cirrus detection from CALIOP and CiPS.

|  |  | CALIOP | |
|---|---|---|---|
|  |  | Cirrus | No cirrus |
| CiPS | Cirrus | $N_{TP}$ | $N_{FP}$ |
|  | No cirrus | $N_{FN}$ | $N_{TN}$ |

## 2.4 Validation metrics

This section introduces the validation metrics used for the validation later on in this paper.

The probability of detection (POD) is used to measure how efficiently CiPS detects cirrus clouds and is given by

$$\text{POD} = \frac{N_{TP}}{N_{TP} + N_{FN}} , \tag{3}$$

where the number of true positives, $N_{TP}$, are all points correctly classified as cirrus and the number of false negatives, $N_{FN}$, all cirrus clouds that remain undetected. The denominator, $N_{TP} + N_{FN}$, is thus the total number of points with a reference cirrus cloud. The false alarm rate (FAR) measures the fraction of cirrus free points that are falsely classified as being cirrus clouds:

$$\text{FAR} = \frac{N_{FP}}{N_{FP} + N_{TN}} . \tag{4}$$

The number of false positives, $N_{FP}$, are all points falsely classified as cirrus (false alarms) and the number of true negatives, $N_{TN}$, all points correctly identified as cirrus free. The denominator, $N_{FP} + N_{TN}$, is thus the total number of points with no reference cirrus cloud. The corresponding CALIOP data are used as a reference when calculating the POD and FAR. Table 1 clarifies the quantities used to calculate the POD and FAR. The POD and FAR are also used to measure how effectively CiPS can determine the opacity/transparency of detected cirrus clouds.

The mean percentage error (MPE) and mean absolute percentage error (MAPE) are used to measure the accuracy of the CTH, IOT and IWP retrievals with respect to CALIOP. The MPE is given by

$$\text{MPE} = \frac{100\%}{N} \sum_{i=1}^{N} \frac{E_i - O_i}{O_i}, \tag{5}$$

where $O$ is the observed value by CALIOP and $E$ the estimated value by CiPS and the sum spans over all samples $i = 1, \ldots, N$ used for the evaluation. The MPE gives information about the direction of the deviations, i.e. whether CiPS tends to overestimate (positive MPE) or underestimate (negative MPE) the values with respect to CALIOP. When calculating the MPE, over- and underestimations can cancel out each other, potentially leading to zero MPE (bias) even if the magnitude of the errors is large. Therefore the MAPE has been considered as well. The MAPE is given by

$$\text{MAPE} = \frac{100\%}{N} \sum_{i=1}^{N} \left| \frac{E_i - O_i}{O_i} \right|, \tag{6}$$

and gives information about the average magnitude of the errors relative to the expected values observed by CALIOP. A vanishing MAPE means no deviation from the observed values and a perfect correlation.

## 2.5 The COCS algorithm

The COCS (Cirrus Optical properties from CALIOP and SEVIRI) algorithm retrieves CTH and IOT of cirrus clouds from SEVIRI (Kox et al., 2014). It combines V2 CALIOP L2 cloud layer data, SEVIRI thermal observations and auxiliary data using an ANN to retrieve CALIOP-like cirrus properties for the full SEVIRI field of view every 15 min and 24 h per day. The cirrus properties retrieved with COCS are used for comparison with CiPS in Sect. 4.2 and COCS is thus shortly introduced here.

COCS is a MLP with ten input neurons (7 brightness temperatures and temperature differences, viewing zenith angle, land-sea mask and latitude), two output neurons (IOT and CTH) and 600 neurons in one single hidden layer. COCS was trained with three years of data including SEVIRI observations from both MSG-1 and MSG-2. The detection of cirrus clouds takes place indirectly in COCS: a pixel is cirrus covered if its IOT ($IOT_{COCS}$) $\geq 0.1$, meaning that pixels with $IOT_{COCS} < 0.1$ are considered too uncertain and regarded as cirrus free. The value of 0.1 was chosen as a trade-off between high POD and low FAR.

The V2 CALIOP L2 cloud layer products contain no information on data quality and the feature classification flag and feature optical thickness among other variables were released as beta products (early release). CALIOP V2 layer data used in Kox et al. (2014) had to fulfil three filtering conditions to be classified as a cirrus cloud: 1) To exclude inaccurate retrievals due to diverging extinction retrievals in opaque cloud layers, the maximum IOT was limited to 2.5. 2) To ensure that the cirrus clouds were not falsely classified layers of aerosols or liquid water clouds, the mid-layer temperature had to be 243 K or colder. 3) The layer top height had to exceed 9.5 km in the tropics and 4.5 km in polar regions, with a linear decrease between these two values in mid-latitudes.

## 3 CiPS

CiPS is the new algorithm, based on the heritage from COCS in the sense that it also utilises artificial neural networks primarily trained with SEVIRI and CALIOP data. Significant enhancements with regards to the ANN structure, training input and output data and training methodology have been implemented, in order to improve on retrieval performance and computational speed. In addition to CTH and IOT, CiPS is also trained to retrieve cloud opacity information and IWP.

### 3.1 Multiple artificial neural networks

In contrast to COCS that uses one single ANN to retrieve IOT and CTH, CiPS utilises four ANNs, making it possible to customise the input variables, training data and ANN structures individually for each task to be solved.

1. The first ANN is a classification network trained to detect cirrus clouds using a binary cirrus cloud flag (CCF). Due to the continuous activation function used by the ANN (Sect. 2.3.1), the retrieved value of the CCF neuron is a real number

in the interval (0,1) represented by a 32-bit floating point number. This value can be interpreted as a cirrus probability, where high and low values indicate a high and low probability of cirrus presence respectively. This provides at least three major advantages over an IOT threshold based detection: 1) The CCF detection threshold (0–1) can be determined depending on the application. A higher threshold means a lower FAR, whereas a lower threshold means a higher POD (Eq. (3)). 2) The cirrus detection is independent of the IOT and not limited to cirrus clouds with an estimated optical thickness greater than 0.1, as is the case for COCS. 3) Since no additional information is needed for the pixels classified as cirrus free by the cirrus detection ANN, the ANNs for CTH, IOT, IWP and opacity information retrieval can be trained only with cases where cirrus clouds are present. This excludes a large number of largely different input data combinations representing the same "cirrus" properties, i.e. the situations where $IOT = IWP = 0$.

2. The second ANN is used for the CTH retrieval.

3. The third ANN is used for the IOT/IWP retrieval. These two variables are provided by the same network since they are physically closely related (Heymsfield et al., 2005).

4. CALIOP cannot provide accurate IOT/IWP retrievals for thicker cirrus clouds where the laser beam is completely attenuated. Hence the estimated IOT and IWP by CiPS for such situations should not be trusted. Therefore a second classification network is introduced with CiPS, trained to identify the cirrus clouds where CALIOP is saturated. Similarly to the cirrus detection ANN, the opacity classification ANN retrieves real numbers in the interval (0,1), that can be regarded as an opacity probability information. From here a binary opacity flag (OPF) is obtained using a suitable opacity classification threshold (Sect. 3.6).

## 3.2 Input data

The following subsections introduce all input data used to train CiPS. An overview is provided in Tab. 2.

### 3.2.1 Brightness temperatures from SEVIRI

Brightness temperatures from all thermal channels of SEVIRI except for the ozone channel at 9.7 μm are used. The brightness temperatures are calculated according to EUMETSAT (2012). The ozone channel is excluded because its sensitivity peaks in the stratosphere, where no cirrus clouds are present, and because of its strong annual cycle due to the ozone variability (Ewald et al., 2013). Channels with significant solar contribution are excluded in order to have the same conditions and similar performance during both day and night. Alongside the single brightness temperatures, CiPS works pixel by pixel and takes advantage of the information from nearby pixels by utilising the regional *maximum* brightness temperatures for the 3 window channels centred at 8.7, 10.8 and 12.0 μm. The regional maximum temperature is identified for each pixel as the maximum temperature within a $19 \times 19$ pixels ($\approx 57 \times 57 \, \text{km}^2$ at nadir) large box centred at the pixel under consideration. The idea with the regional maximum brightness temperature is to estimate the temperature that SEVIRI would observe for a cirrus covered pixel if the pixel was cirrus free. This is done by assuming that at least one of the 361 pixels within the box is not covered by

a cirrus cloud (Krebs et al., 2007). The corresponding cirrus free temperature is useful both for the detection of cirrus clouds and the retrieval of the cirrus properties since it provides information about the up-welling radiation from the surface or lower water clouds. The box size of $19 \times 19$ pixels is chosen such that the region is small enough to reduce the risk of unrepresentative maximum temperatures over inhomogeneous surfaces (e.g. coast lines), but large enough to increase the chance of capturing a representative cirrus free pixel.

For the classification ANNs (CCF and OPF) the regional *average* brightness temperatures for the 2 water vapour channels centred at 6.2 and 7.3 μm are used as well. The regional averaged brightness temperatures are calculated for each pixel as the boxcar average temperature within a $19 \times 19$ pixels large box centred at the pixel under consideration. A homogeneous area with cold temperatures indicates the presence of a thick cirrus cloud. The combination of a single temperature and the corresponding regional average for the water vapour channels provides information about high cloud structures useful for the detection of cirrus clouds (Krebs et al., 2007).

### 3.2.2 Surface temperature from ECMWF

With CiPS we introduce modelled data from the ECMWF ERA-Interim re-analysis dataset (Dee et al., 2011) to the list of input variables. The surface skin temperature $T_{\mathrm{surf}}$ is strongly related to the thermal radiation emitted by the Earth and thus related to the brightness temperatures observed by SEVIRI. This information helps accounting for the radiation emitted by the surface which is partly transmitted in the satellite direction through thin cirrus. It also helps the ANNs to distinguish between cirrus clouds and cold surfaces like Greenland and Antarctica. The temporal resolution of 6 h and spatial resolution of 0.125°is used.

### 3.2.3 Auxiliary data

Along with the data provided by SEVIRI and ECMWF, additional auxiliary datasets are used. The latitude provides valuable information about the geographical location with respect to the global circulation convergence and divergence zones (e.g. the ITCZ, subsidence regions and the polar front) which strongly affect the presence and properties of cirrus clouds. Considering the SEVIRI viewing zenith angle, the SEVIRI pixel size and slant path length are implicitly accounted for. Two flags indicating the presence of surface water and permanent ice/snow respectively are included to gain additional information about the observed surface type. Due to the seasonal variations in the global circulation and the presence of cirrus clouds (Stubenrauch et al., 2013) the day of the year (DOY) is used. The DOY is converted to two variables, $\sin(2\pi\,\mathrm{DOY}/365)$ and $\cos(2\pi\,\mathrm{DOY}/365)$ in order to remove the hard transition from December 31 to January 1. Two variables are used to avoid the repeating pattern of sine or cosine alone.

### 3.3 Output data: cirrus properties from CALIOP

The cirrus presence and properties, including a CCF and an OPF as well as the CTH, IOT and IWP are derived from the V3 CALIOP L2 5 km cloud and aerosol layer products (CAL_LID_L2_05kmC|ALay-Prov-V3-0X..., CALIPSO Science Team, 2016a, b). Major improvements with respect to V2 data include enhanced cloud-aerosol discrimination, improved cloud ther-

**Table 2.** Input data used to train the four ANNs contained in CiPS. BT = Brightness Temperature, regavg = regional average, regmax = regional maximum, VZA = viewing zenith angle

| | CCF | OPF | CTH | IOT/IWP |
|---|---|---|---|---|
| $BT_{6.2\mu m}$ | ✓ | ✓ | ✓ | ✓ |
| $BT_{7.3\mu m}$ | ✓ | ✓ | ✓ | ✓ |
| $BT_{8.7\mu m}$ | ✓ | ✓ | ✓ | ✓ |
| $BT_{10.8\mu m}$ | ✓ | ✓ | ✓ | ✓ |
| $BT_{12.0\mu m}$ | ✓ | ✓ | ✓ | ✓ |
| $BT_{13.4\mu m}$ | ✓ | ✓ | ✓ | ✓ |
| $BT_{6.2\mu m, regavg}$ | ✓ | ✓ | | |
| $BT_{7.3\mu m, regavg}$ | ✓ | ✓ | | |
| $BT_{8.7\mu m, regmax}$ | ✓ | ✓ | ✓ | ✓ |
| $BT_{10.8\mu m, regmax}$ | ✓ | ✓ | ✓ | ✓ |
| $BT_{12.0\mu m, regmax}$ | ✓ | ✓ | ✓ | ✓ |
| $T_{surf}$ | ✓ | ✓ | ✓ | ✓ |
| Latitude | ✓ | ✓ | ✓ | ✓ |
| VZA | ✓ | ✓ | ✓ | ✓ |
| Water flag | ✓ | ✓ | ✓ | ✓ |
| Snow/ice flag | ✓ | ✓ | ✓ | ✓ |
| $\sin(2\pi\frac{DOY}{365})$ | ✓ | ✓ | ✓ | ✓ |
| $\cos(2\pi\frac{DOY}{365})$ | ✓ | ✓ | ✓ | ✓ |

modynamic phase determination, more accurate estimates of layer spatial and optical properties as well as an improved estimate of the low cloud fraction. Furthermore, new products like the IWP and retrieval uncertainties are included. Most importantly, the maturity level of all products used to develop CiPS have been upgraded from beta status to provisional or higher, meaning that the data have at least been compared to independent sources in order to correct obvious artefacts (NASA Atmospheric
5  Science Data Center, 2010).

Even though the cloud and aerosol layer product are reported with a spatial resolution of 5 km, two additional coarser resolutions of 20 and 80 km are used to detect the cloud and aerosol layers reported in the 5 km products (Vaughan et al., 2009). At a spatial resolution of 5 km, the signal-to-noise ratio of a faint cirrus or aerosol layer is usually too weak to be distinguished from the clear-sky atmospheric signal. By averaging 4 or 16 consecutive 5 km profiles the signal-to-noise ratio is increased,
10  which allows for detection of very thin cirrus and aerosol layers. For example if a thin cirrus cloud with an optical thickness of 0.1 and a top altitude of 10 km is identified only when 16 consecutive 5 km profiles are averaged (80 km spatial resolution), 16 consecutive bins in the L2 5 km cloud layer data will report an optical thickness of 0.1 and a top altitude of 10 km. This can result in a vertical overlap between layers detected at different spatial resolutions. This is accounted for by identifying the

part of an icy layer vertically overlapped by another layer (water cloud or aerosol) detected at a higher spatial resolution and correcting the corresponding extinction coefficients, ice water content and CTH accordingly. The column IOT and IWP are then derived by combining the properties of all icy layers in each profile. Finally, the OPF is extracted from the 'Opacity_Flag' product. The Opacity_Flag gives the information whether the CALIOP backscatter signal was completely attenuated within a
detected layer. During the CALIOP retrieval, a cirrus cloud layer is classified as opaque if it is the lowermost layer and not identified as a surface return (Vaughan et al., 2005). A digital elevation model is partly used to identify surface returns, meaning that high cirrus clouds should not be falsely classified with respect to transparency. Cirrus cloud layers detected at the coarser 20 km or 80 km resolutions are classified as transparent if the corresponding base altitude is higher than the lowermost detected feature in at least 50 % of the 4 or 16 consecutive 5 km profiles that constitute the 20 km and 80 km averages.

The minimum detectable backscatter of CALIOP depends on the scattering target (the cirrus cloud in this case), the altitude as well as the vertical and horizontal averaging of the data (McGill et al., 2007). Davis et al. (2010) show that CALIOP can detect approx. one third of the sub-visual cirrus clouds with an optical thickness below 0.01.

The improved quality of the V3 CALIOP products allows us to omit the filtering processes applied to the V2 data used for COCS (see Sect. 2.5). To assure a high quality dataset, the extinction quality control flag, retrieval uncertainties and the
feature classification flag including the quality assessments have been considered. All columns containing at least one layer with unknown feature type, unknown cloud phase or a feature/phase quality assessment flag less than 3 (high confidence) are excluded. Additionally, only those columns with solely constrained or unconstrained cirrus/ice cloud retrievals where the initial lidar ratio remained unchanged during the solution process are included. Furthermore, the columns containing stratospheric features are excluded due to lack of information about whether the features are stratospheric clouds or aerosol layers.

In the following, all quantities referring to CALIOP will be denoted as $IOT_{CALIOP}$, $IWP_{CALIOP}$ and $CTH_{CALIOP}$.

The CALIOP products are chosen as training reference data for CiPS as they should provide the most accurate estimates of especially CTH but also IOT for thin cirrus clouds from space. It is important to note that an ANN can never be better than its training reference and all deficiencies and/or biases in the training reference data will be inherited by the ANN. Since possibly inherited artefacts of the ANN will not show when validated against independent CALIOP retrievals, one must be aware of the
accuracy and limitations of the training data.

Yorks et al. (2011) and Hlavka et al. (2012) validate the spatial and optical properties of cirrus clouds from the V3 CALIOP products using the airborne Cloud Physics Lidar (CPL, McGill et al. (2002)) during the CALIPSO-CloudSat Validation Experiment (CC-VEX). CPL has a higher signal-to-noise ratio (SNR), higher vertical and horizontal resolution and lower multiple scattering compared to CALIOP, making it the most comprehensive tool for validating the CALIOP retrieved cirrus properties.
Ten underpass flights with CALIOP were performed and over 9 500 bins of collocated extinction coefficients were obtained. During the ten flights, extinction coefficients ranging from approx. $0.001–10 \, km^{-1}$ and column optical thickness up to approx. 3 were retrieved. CALIOP and CPL agree on 90 % of the scene classifications (cirrus or no cirrus) on average. For all bins classified as cirrus by CPL, CALIOP agrees on 82 % and for the bins classified as cirrus free by CPL, CALIOP agrees on 91 %. For cases where both CALIOP and CPR detect cirrus, the agreement in cirrus top height is excellent (Yorks et al., 2011).

For transparent cirrus layers the agreement in IOT between CALIOP and CPL is good with on average 15 % higher extinction for CALIOP (0.65 in correlation between CALIOP and CPL). For the unconstrained retrievals where the initial lidar ratio remains unchanged the average difference in extinction is only 7 % (0.80 in correlation between CALIOP and CPL Hlavka et al., 2012). The latter are the ones used to train CiPS (see above), along with the constrained retrievals. At the time of the CC-VEX campaign (between July 26 and August 14 2006) the laser of CALIOP was pointing just 0.3 degrees from nadir leading to a strong specular reflection by layers of horizontally orientated ice (HOI) (Winker et al., 2009). This lead to disagreements in the extinction retrieval with CPL, whose laser pointed 2 degrees from nadir and therefore only received a very small fraction of specular reflections from the HOI (Hlavka et al., 2012). Since November 2007 the CALIOP lidar points three degrees from nadir in order to overcome this issue for layers with HOI. When the column optical thickness is derived for all cirrus covered bins, the relative difference between CALIOP and CPL is only 2.2 % due to cancellation of opposing CALIOP effects. Holz et al. (2016) recently showed that the single layer IOT derived from unconstrained CALIOP retrievals is low-biased with respect to a single channel thermal/IR IOT retrieval combining CALIOP/MODIS observations and forward radiative transfer modelling. The bias is shown to increase with increasing IOT.

The accuracy of the CALIOP IWC/IWP is directly related to the accuracy of the extinction retrievals as well as the IWC parametrization from Heymsfield et al. (2005). A proper independent validation of the CALIOP IWC/IWP is a difficult task due to the lack of reference data at a comparable spatial and temporal resolution. Protat et al. (2010) evaluate the IWC parametrization used for CALIOP for tropical cirrus using ground based radar-lidar retrievals. The results suggest that the parametrization is quite robust and is shown to work well at most altitudes. Above $\sim$12 km the IWC is clearly underestimated with respect to the ground based radar-lidar retrieval. Avery et al. (2012) evaluate the CALIOP IWC using coincident data from CloudSat and in situ measurements inside a tropical convective cloud. At the lower altitudes (8–12 km), the CALIOP IWC is underestimated with respect to the in situ measurements, which could be attributed to a lower penetration depth of CALIOP and the removal of CALIOP layers containing HOI. Between 12–14 km the agreement between the CALIOP IWC and the in situ measurements is good. At all altitudes CALIOP seems to underestimate the IWC with respect to CloudSat. Wu et al. (2014) show that the V3 CALIOP IWC agrees well with airborne in-situ measurements up to approx. 20 mg m$^{-3}$ at an altitude of 12 km. The CALIOP IWC agrees well with the CloudSat IWC within the regions where their sensitivities overlap. This occurs between 5–20 mg m$^{-3}$ at an altitude of 12 km and between 30–200 mg m$^{-3}$ at 15 km.

## 3.4 Data preparation

To learn the relationship between the SEVIRI, ECMWF, auxiliary data and the cirrus properties from CALIOP, an extensive dataset is created containing spatial and temporal collocations of all variables. The training dataset covers the time period from April 2007 to January 2013, which is the time when MSG-2 was the operational satellite at 0.0° E. CiPS is restricted to MSG-2 alone, since we did not want to mix data from multiple SEVIRI instruments since their characteristics are slightly different.

### 3.4.1 Data collocation

For this time period all quality controlled CALIOP data within the SEVIRI field of view are identified and collocated with single SEVIRI pixels in time and space. Due to the different viewing geometries of SEVIRI and CALIOP, the same cloud seen by SEVIRI and CALIOP at the same time appears to be located at two different positions. The magnitude of this displacement depends on the viewing angle and the altitude of the cloud layer. This effect has been corrected for using the latitude, longitude and cloud top altitude from CALIOP (parallax correction) to project ice clouds to the SEVIRI grid. The cirrus properties from CALIOP are spatially collocated with SEVIRI observations from the pixel having the largest overlap with the 5 km CALIOP orbit segment. The data are temporally collocated by identifying the SEVIRI observation that has the smallest difference in acquisition time compared to CALIOP. With a temporal resolution of 15 min for SEVIRI, the maximum difference in acquisition time between SEVIRI and CALIOP is 7.5 min.

When collocating SEVIRI and CALIOP observations with the purpose of training an ANN one must consider two aspects. 1) Even though the 5 km average of CALIOP point measurements fits the spatial resolution of SEVIRI ($3 \times 3\,km^2$ at nadir and approx. $4 \times 5\,km^2$ in mid-latitudes) quite well in the along-track direction, the two observations differ largely in scale in the across-track direction as the footprint of CALIOP is approx. 70 m wide at the Earth's surface. Consequently the 5 km CALIOP orbit segment is representative only for a relatively small fraction of a SEVIRI pixel. This will induce inevitable errors and lead to imperfect information used to train the ANN. This is especially relevant for partial cloud cover, where CALIOP may observe a cloud free area in an otherwise cloud covered SEVIRI pixel. If the error from imperfect collocations is random, this will have a limited effect on the ANN. Only if there is a recurrent systematic difference as a result of the different spatial scales this will be lead to biased retrievals (Holl et al., 2014). 2) Although cirrus clouds leave their mark on both SEVIRI and CALIOP measurements in a similar way, SEVIRI does not share CALIOP's possibility of discerning vertically separated ice clouds, liquid water clouds and aerosols. Consequently SEVIRI should not be expected to discern the signal from liquid water clouds and aerosols when retrieving the IOT as effectively as CALIOP.

The ECMWF surface temperatures are spatially collocated with the satellite observations using nearest neighbour. For the temporal collocation, the ECMWF re-analysis data are linearly interpolated between the ECMWF 6 h time steps and the satellite acquisition time.

### 3.4.2 Training and validation data

The full collocated dataset, covering the entire SEVIRI disc and a time period of almost 6 years, contains close to 50 million collocations. 80 % of those collocations are used to create the four datasets required for the training of the four ANNs contained in CiPS. For the CCF ANN, both cirrus free collocations as well as collocations with transparent and opaque cirrus clouds are included in the training dataset. Collocations with no cirrus cloud present are excluded from the training datasets used to train the OPF ANN as well as the CTH and IOT/IWP retrieval ANNs, since those networks will be applied only on pixels identified as cirrus covered by the CCF ANN. Furthermore, the IOT/IWP ANN is trained only with collocations containing transparent cirrus clouds, where the CALIOP signal was not saturated such that the true, rather than the apparent, IOT and IWP could

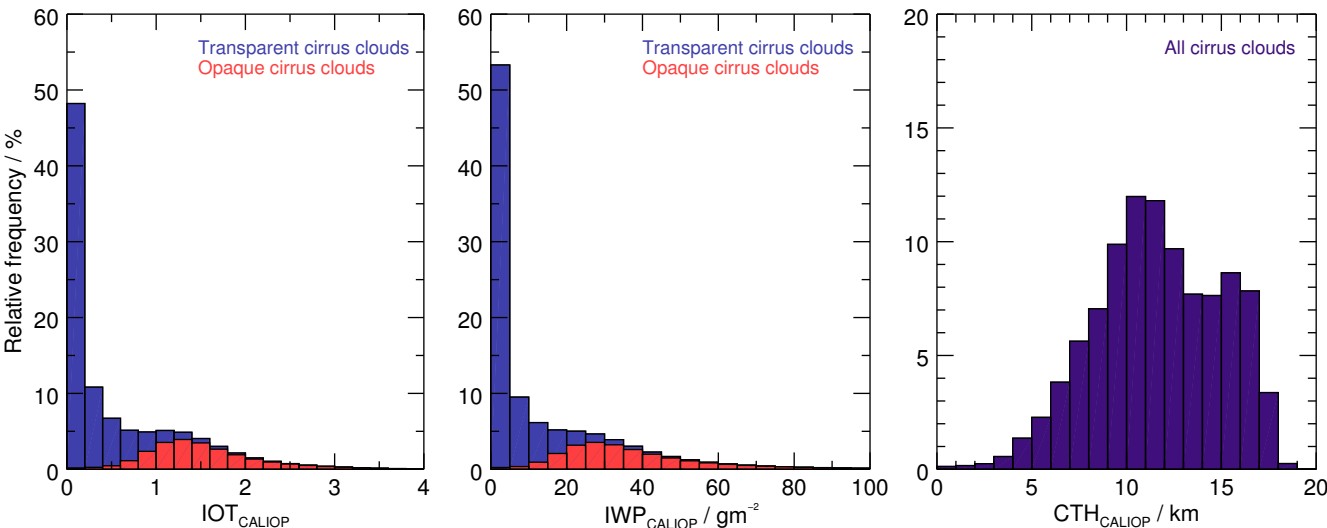

**Figure 2.** The relative number distribution of the cirrus IOT (bin size $= 0.2$), IWP (bin size $= 5\,\mathrm{g\,m^{-2}}$) and CTH (bin size $= 1\,\mathrm{km}$), from almost 6 years of V3 CALIOP L2 layer data over the SEVIRI disc.

be retrieved. Figure 2 shows the relative number distributions of the CTH, IOT and IWP retrieved by CALIOP. It is clear that the collocation dataset is unbalanced in several aspects. The IOT and IWP have exponential distributions with very few thicker cirrus clouds. Similarly there are comparably few low and high cirrus clouds available and the CTH distribution has two peaks, corresponding to mid-latitudes and tropics. To improve the end performance for those rare points the unbalance

of the training datasets is mildered "by hand". For the cirrus detection and IOT/IWP ANNs, four duplicates of all cirrus clouds with $\mathrm{IOT_{CALIOP}} \geq 1.0$ have been added to the training datasets. Similarly four duplicates of all cirrus clouds with $\mathrm{CTH_{CALIOP}} > 17\,\mathrm{km}$ or $\mathrm{CTH_{CALIOP}} < 5\,\mathrm{km}$ have been added to the CTH training dataset. For the opacity classification ANN, four duplicates of all opaque cirrus clouds have been added to the training dataset. This approach does not introduce any new information that the ANNs can learn from, but increases the weight of the added points during the training. Adding too few

duplicates has a negligible effect whereas too many duplicates give the added points a too strong impact during the training. By testing different numbers, four duplicates are seen to yield the best results for all ANNs. Furthermore, the IOT and IWP are transformed to their logarithmic counterparts before the training ($\mathrm{IOT}^{*} = \log_{10}(\mathrm{IOT}), \mathrm{IWP}^{*} = \log_{10}(\mathrm{IWP}/1\,\mathrm{g\,m^{-2}})$). Finally the single input variables are normalised to have zero mean and unit variance (LeCun et al., 1998) and the output data are scaled to fit the ranges of the activation functions (Sect. 3.5) used by the ANNs.

The remaining 20 % of the collocation dataset is used for validation. Half of this data is used to create *internal validation* datasets that are used to monitor the error against independent data during the training in order to avoid overfitting (see Sect. 3.5) and to determine training meta-parameters, ANN structures (see Sect. 3.7) and classification thresholds (see Sect. 3.6). The internal validation datasets have been filtered in the same manner as the training datasets, but have not been balanced by adding duplicates of selected points. The second half of the validation data is used for the *final validation* of CiPS (and COCS)

presented in Sect. 4.2. This final validation data are not used for any purpose during the development and training of CiPS. With common ANN terminology the internal and final validation data are usually referred to as validation and test data respectively.

## 3.5 Training

To train and apply CiPS the Fast Artificial Neural Network library (FANN, Nissen, 2003) is used. The four ANNs contained in CiPS are trained using the standard backpropagation algorithm and mini-batch gradient descent described in Sect. 2.3.2.

Three hidden layers are used for the cirrus cloud detection, two for the CTH and IOT & IWP retrievals and a single hidden layer for the opacity classification. All ANNs use 16 hidden neurons per hidden layer (see Sect. 3.7 for details on the MLP structures). For the classification ANNs (CCF, OPF) the sigmoid activation function is used for both hidden and output layers, whereas the tanh activation function is used for hidden and output layers for the regression ANNs (CTH, IOT & IWP). A batch size of 1024 is used, meaning that the ANNs look at 1024 input and output data combinations before each weight update. The value of 1024 was chosen as a trade-off between the noise in the error gradient that increases with smaller batch sizes and the required computational power that increases with larger batch sizes. The learning rate and momentum are sensitive to the problem that should be solved, the corresponding training data as well as the number of input and output variables (Schaul et al., 2013). To find the optimal values an extensive iterative test approach is performed. For this test a large GPU cluster (120 TFLOPs - 20 NVIDIA GTX Titan GPUs) is used to train numerous ANNs with different numbers of hidden layers and hidden neurons and a wide range of learning rates and momentum values. To find the optimal values for each meta-parameter, a random search according to Bergstra and Bengio (2012) is performed within intervals chosen based on expert knowledge. Sets of meta-parameters are randomly drawn from the pre-defined intervals and used to train corresponding sets of ANNs. Assuming an infinite number of samples, this procedure can be regarded as a global optimization technique. The optimal set of meta-parameters is defined as the one that minimises the mean squared error (MSE) between the ANN and the internal validation data. All resulting optima are well within these chosen intervals, so it is assumed that the choice of the intervals does not introduce any distortion or bias. For both the classification and regression tasks a learning rate around 0.05 and momentum around 0.99 is found to provide ANNs with the lowest MSE against the independent internal validation data.

The ANNs are initially trained using 25 % the training data. This is done in order to speed up the training. This first phase continues until the accuracy of the ANNs does no longer improve with respect to the internal validation data. During this first phase of the training a rough estimate of the error gradient is sufficient as we are interested in the general direction towards a minimum solution. Thus a larger learning rate and smaller mini-batches are preferred. When the ANN approaches the region of an optimal solution, those large step-sizes and small mini-batches are too blunt to find the finer structures needed to solve the problem better. Thus the learning rate and batch size should be adjusted accordingly in order to make smaller and more informed steps in the search space. During this iterative tuning phase, the learning rate is reduced by a factor 2 and the batch size is increased by a factor 2. In order to not impede the effect of the finer learning rate and batch size, the momentum is reduced accordingly. Furthermore the size of the training dataset, which started at 25 % during the first phase, is increased by a factor 2. This is a schedule procedure that is commonly used in the Machine Learning/ANN community. As the tuning phase continues the meta-parameters are refined according to the schedule above as soon as the total error stops to decrease

with respect to the internal validation dataset. The tuning phase and thereby the training stops when the respective ANNs have reached a point where additional epochs do not reduce the error, using 100 % of the respective training datasets.

To avoid over-fitting, the error against the independent internal validation datasets (Sect. 3.4.2) is always monitored. Over-fitting occurs when an ANN learns the training dataset itself rather than the relationship between the input and output variables and thus loses its ability to generalise. To make sure that the ANNs are not over-fitting the updated weights are only saved if the error against the internal validation dataset decreases, otherwise the training continues but the set of weights having the current minimum error against the internal validation dataset is kept.

For each task/ANN the training is repeated twice in order to reduce the risk of having a bad end performance as a result of a bad set of initial weights (from Widrow and Nguyen's algorithm (Nguyen and Widrow, 1990)). In the end, only the best performing network is used. The differences between the two networks trained for each task/ANN are however very small (ca. 3 ‰ relative difference in MSE).

Using a common standard desktop PC (using 1 core à 3.40 GHz, Intel Core i5-3570), the final set of ANNs, which we call CiPS (Cirrus Properties from SEVIRI), takes approx. 60 seconds to process a complete SEVIRI image (3712×3712 pixels) including I/O. Ca. 40 seconds are needed for the cirrus cloud detection and another 20-30 seconds for the opacity classification as well as the retrieval of CTH, IOT and IWP. The cirrus cloud detection takes longer as this ANN is applied to all SEVIRI pixels, whereas the other ANNs are only applied on those pixels classified as icy by CiPS. This is ca. 10 times faster than the combined CTH and IOT retrieval by COCS (Kox et al., 2014). ANN computations are highly parallelizable, meaning that the computation time can be reduced significantly by distributing the computations across multiple cores.

### 3.6 Cirrus detection and opacity classification thresholds

As described in Sect. 3.1 the thresholds for the CiPS CCF and OPF ANNs can be selected between 0–1 depending on the application. These two thresholds are chosen based on a trade-off between the POD (Eq. (3)) and FAR (Eq. (4)) using the internal validation dataset. Figure 3 shows the FAR and POD of the CiPS classification ANNs as a function of classification threshold (also known as the receiver operating characteristic (ROC) curve). It is clear that the two quantities are anti-correlated where a lower threshold yields a higher POD, but comes at the expense of an increased FAR and vice versa. For the application and validation presented in Sect. 4 and 5 as well as for the standard usage of CiPS, a CCF threshold of 0.62 is chosen, resulting in a total POD of 71 % and a FAR of 3.9 %. The low POD is a direct effect of the large amount of very thin to sub-visual cirrus (IOT $< 0.03$) that CiPS does not detect (see Fig. 2 and 7). For the OPF a threshold of 0.86 is chosen, resulting in a POD of 71 % and a FAR of 4.0 % for the cirrus clouds that CiPS successfully detects. The two thresholds chosen for CiPS are indicated in Fig. 3 with red circles.

### 3.7 Evaluation of different MLP structures

When developing CiPS, several ANNs with different MLP structures were trained in order to investigate the effect of the MLP structure on the end performance and to determine the respective structures that offer the best trade-off between accuracy and application time. For each ANN contained in CiPS several networks with different structures were trained using one, two and

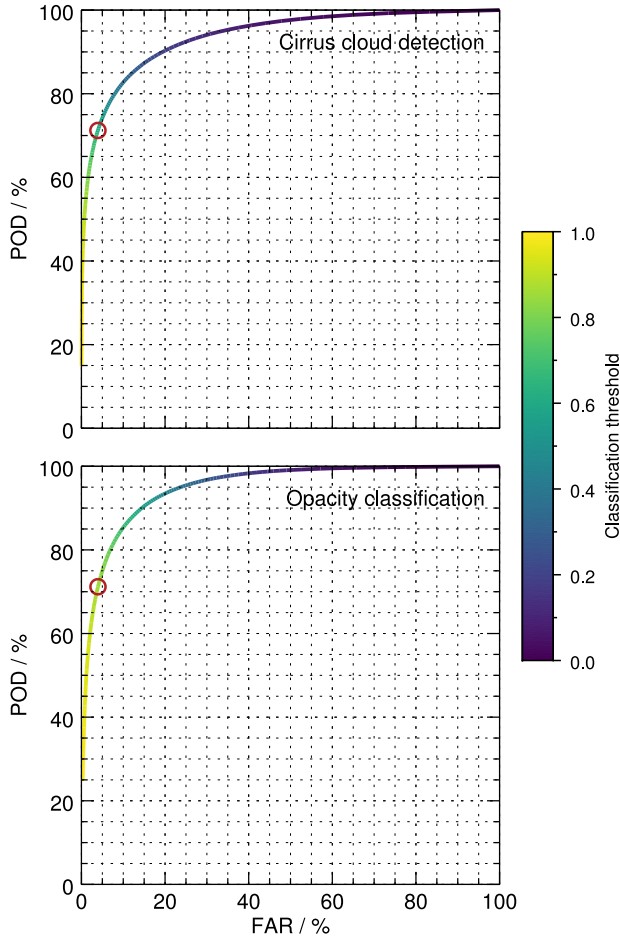

**Figure 3.** The POD and FAR of the CiPS cirrus cloud detection and opacity classification ANNs as a function of classification threshold. The red circles indicate the final thresholds selected for the two ANNs.

three hidden layers with either 16 or 64 hidden neurons per hidden layer. For the single hidden layer structures we also train with 128 hidden neurons. Also here the training was repeated twice for each network in order to reduce the risk of having a bad end performance as a result of a bad set of initial weights. Again, only the best performing network among the two is further evaluated after the training. All different structures were trained according to the *first phase* as explained above (Sect. 3.5) i.e.

5 using 25 % of the respective datasets. After this stage the accuracy of the different MLP structures was evaluated and compared using the internal validation datasets. This investigation was used to determine the MLP structures used for CiPS (see Sect. 3.5).

Figure 4a shows the difference in POD (Eq. (3)) between each structure and the least complex structure having one hidden layer and 16 hidden neurons (denoted as 1-16) for the cirrus cloud detection ANN with respect to CALIOP for the seven different structures that were investigated. Similarly, Fig. 4b and Fig. 4c show the difference in MAPE (Eq. (6)) between each

10 structure and the least complex one for the CTH and IOT retrieval ANNs respectively. The MAPE behaviour of the IWP is very

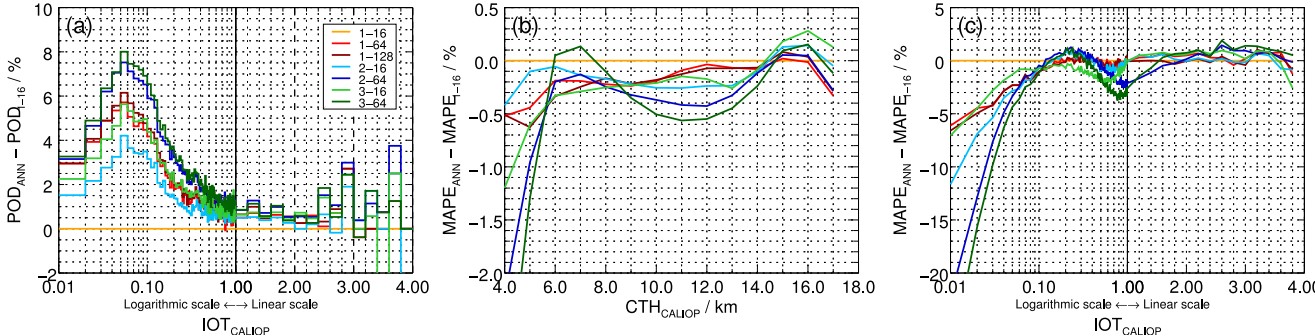

**Figure 4.** The difference in accuracy between each MLP structure and the least complex MLP structure having one hidden layer with 16 hidden neurons (1-16). (a) The difference in POD for the cirrus cloud detection, (b) the difference in MAPE for the CTH retrieval and (c) the difference in MAPE for the IOT retrieval. The number to the left of the hyphen is the number of hidden layers and the number to the right the number of hidden neurons per hidden layer.

**Table 3.** Approximate time required to process 1 million data points using the different ANN structures investigated in this study. The number to the left of the hyphen is the number of hidden layers and the number to the right the number of hidden neurons per hidden layer.

| Structure | 1-16 | 2-16 | 3-16 | 1-64 | 1-128 | 2-64 | 3-64 |
|---|---|---|---|---|---|---|---|
| Time / s | 2.1 | 3.1 | 4.0 | 5.2 | 9.5 | 14.4 | 23.6 |

similar to the MAPE of the IOT and is therefore not presented here. For the OPF, the structure of the network does not seem to have any significant influence on the performance and is thus not presented here. For the cirrus detection and IOT retrieval, only the transparent cirrus clouds are considered. Please note that for a better visualisation for the lower IOT values, the horizontal axes in Fig. 4a and 4c are divided into one logarithmic range ($IOT_{CALIOP} < 1.0$) and one linear range ($IOT_{CALIOP} \geq 1.0$).

Furthermore, Table 3 lists the approximate amount of time required to process 1 million data points/pixels (including I/O) using the above specified desktop PC with the different structures.

In all cases, already small networks produce reasonable results. In many cases differences between structures are not very large. Nevertheless, we also see that larger ANNs can always solve the problems in a more accurate way and especially for the cirrus cloud detection it is beneficial to either use more hidden neurons or add more hidden layers rather than using a simple

structure with one hidden layer and 16 hidden neurons (1-16). Using two or three hidden layers with 64 hidden neurons each (2-64, 3-64) yields a POD that is up to 8 percentage points higher compared to one hidden layer with 16 hidden neurons (1-16). Similarly, a structure with three hidden layers and 16 hidden neurons (3-16) yields a POD that is up to 5.5 percentage points higher compared to the structure with one hidden layer and 16 hidden neurons (1-16). Although three hidden layers with 64 neurons each (3-64) offers the highest accuracy for all cases, such a complex structure processes the data significantly slower

by a factor 8 or 6 compared to the smaller structures with 2 or 3 hidden layers and 16 neurons per layer. For the IOT retrieval, a larger ANN is mostly beneficial for the thinner cirrus and the MAPE with respect to CALIOP seems to be saturated and hardly

improvable for $IOT_{CALIOP} > 0.1$ using this approach and training data. For the sub-visual cirrus, the MAPE with respect to the CALIOP reference IOT is up to 13 percentage points lower using two hidden layers instead of one hidden layer with 16 hidden neurons each. For the CTH retrieval, only marginal improvements in the MAPE with respect to CALIOP ($\approx 0.1 - 0.5$ percentage points) are observed using the more complex structures in comparison to the least complex one (1-16). Only for the
lowermost clouds ($CTH_{CALIOP} < 6.0\,km$) the advantage of using more hidden layers and neurons is more evident.

## 4   CiPS application and validation

### 4.1   Application

In this section CiPS is applied to the June 1, 2015 12:30 UTC MSG-3/SEVIRI image subset consisting of $350 \times 350$ pixels comprising western/central Europe. Figure 5a shows a false colour RGB composite that uses three SEVIRI channels centred
at 0.6, 0.8 and 10.8 µm. With this channel combination the thick and thin cirrus clouds are identified as white and blueish respectively, whereas the liquid water clouds are recognised as yellow. Quite intuitively surface water and land appear as dark blue and green respectively. Two large cirrus clouds can be seen ranging from the south-western parts of France towards the Alps and southern parts of Scandinavia respectively. Also over England and Norway, cirrus clouds are present and clearly visible in the RGB. Liquid water clouds are mainly present over the central parts of France, Switzerland and Germany, but also
over the North Sea, Mediterranean Sea and southern parts of Scandinavia. For an enhanced view of thin cirrus clouds Fig. 5b shows the brightness temperatures difference between the SEVIRI channels centred at 8.7 µm and 10.8 µm. In this picture, cirrus clouds are characterised by positive or slightly negative values.

Figure 5c shows the cirrus cloud mask retrieved by CiPS for the same scene. The blue and grey areas show all pixels that CiPS classify as cirrus, of which the grey pixels are classified as opaque. This means that for the grey pixels the retrieved IOT
and IWP is more likely to be underestimated. CiPS clearly detects all cirrus clouds that can be identified in the false colour RGB composite (Fig. 5a) and from the brightness temperature differences (Fig. 5b). The OPF correlates well with the cirrus brightness in the RGB. The brightest parts of the cirrus clouds, which represent the thickest parts, are classified as opaque by CiPS.

Figures 5d-f show the corresponding CTH, IOT and IWP retrieved by CiPS. CiPS captures the latitude dependency of the
CTH, with generally lower values at higher latitudes. We also see elevated heights for the thicker cirrus cloud areas. The cloud edges are generally seen to have lower altitudes, which could indicate ice crystal sedimentation or partial cloud cover inside the SEVIRI pixels. As expected, the IOT and IWP are well correlated and qualitatively the values corresponds well to the level of transparency of the different cirrus clouds seen in Fig. 5a. For a quantitative evaluation of the IOT and IWP as well as the other quantities, readers are referred to Sect. 4.2.

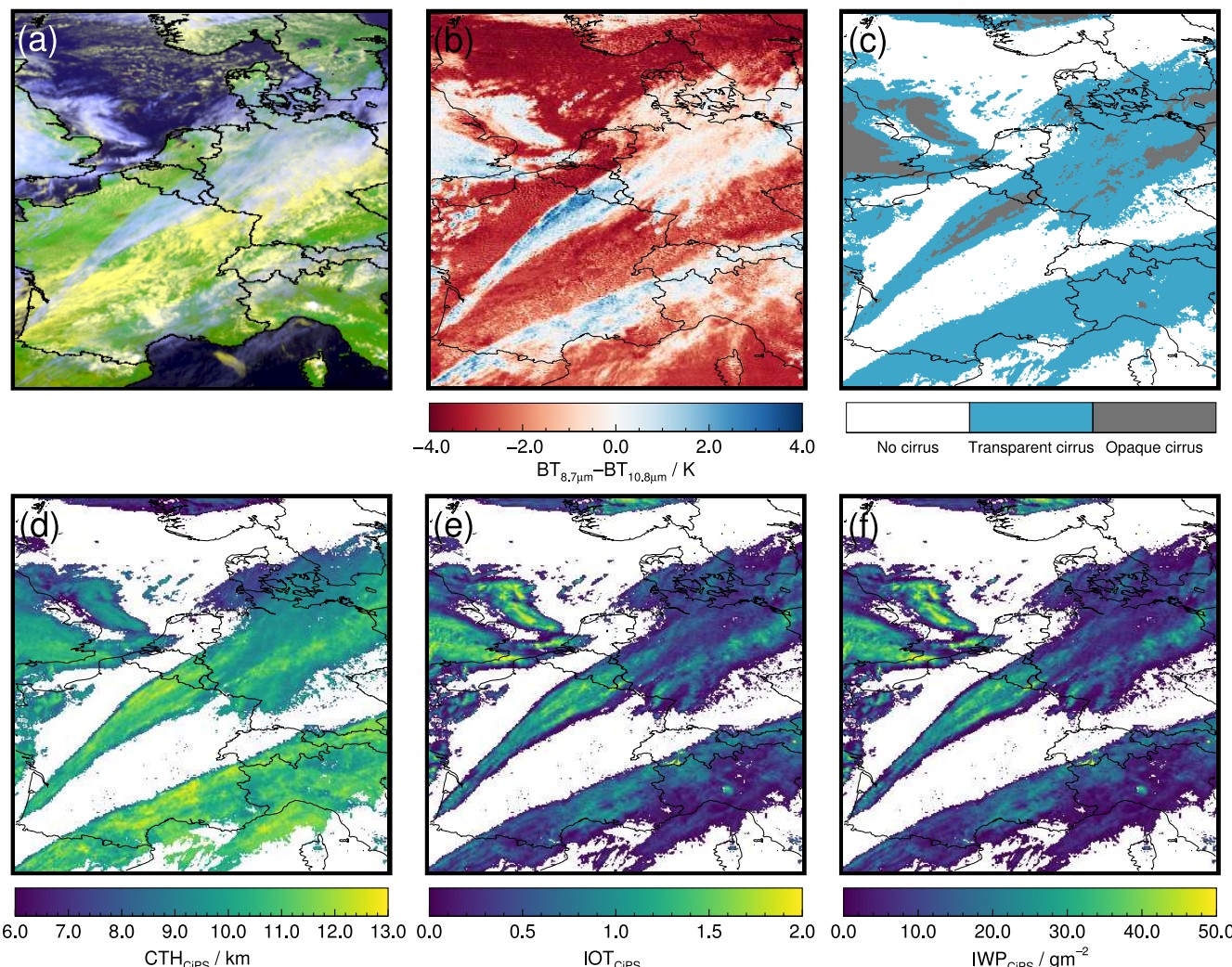

**Figure 5.** (a) MSG-3/SEVIRI false colour RGB composite over parts of Europe on June 1$^{st}$, 2015 at 12.30 UTC and the corresponding (b) brightness temperature difference $BT_{8.7\mu m}$-$BT_{10.8\mu m}$ (c) cirrus cloud mask with opacity information, (d) CTH, (e) IOT and (f) IWP retrieved by CiPS.

## 4.2 Validation against CALIOP

In this section the performance of CiPS is validated against V3 CALIOP products using the 10 % subset (approx. 4.9 millions collocations) of the full collocation dataset excluded from the training of CiPS (Sect. 3.4.2). The results are presented for the full SEVIRI field of view. Since CiPS and COCS share the concept of using ANNs trained with primarily SEVIRI and CALIOP data, we also present the corresponding validation results of COCS. This clarifies the improvements of CiPS compared to COCS.

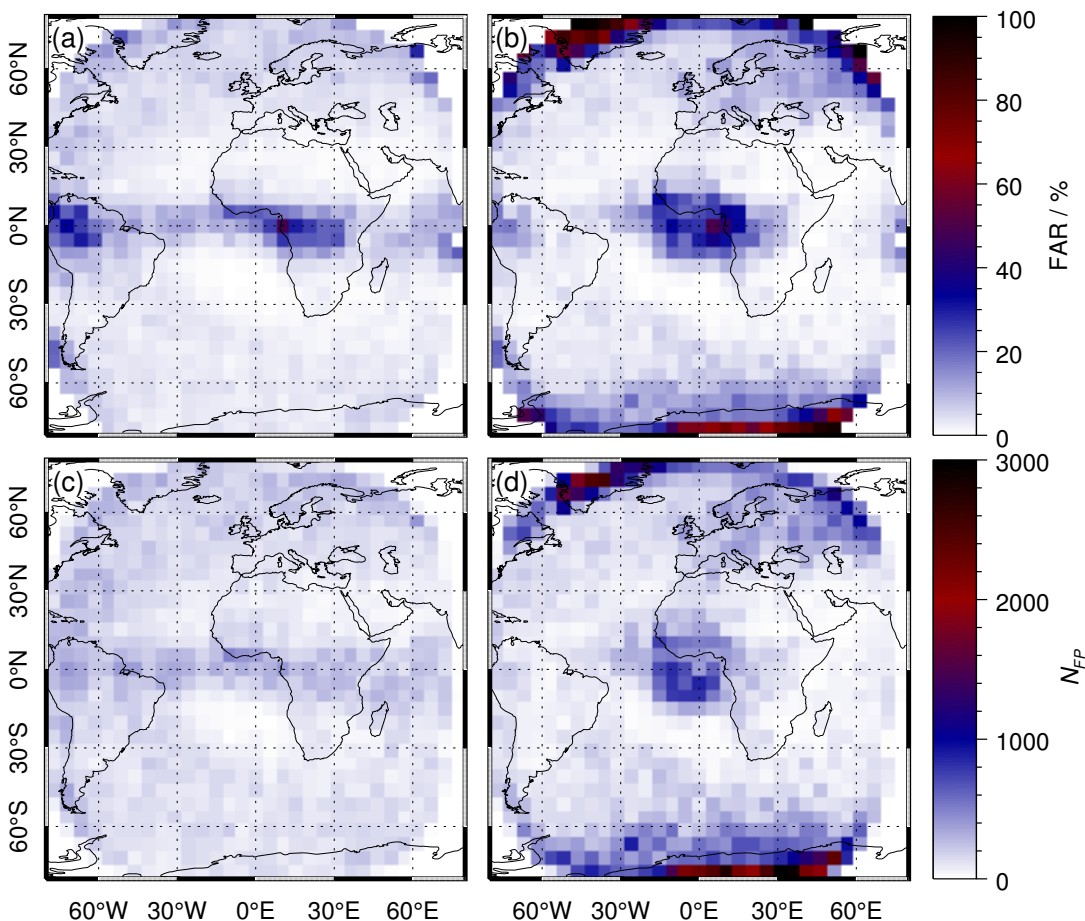

**Figure 6.** Top: the FAR of the CCF retrieved by CiPS (a) and COCS (b). Bottom: the absolute number of false alarms by CiPS (c) and COCS (d). Approx. 3.3 millions cirrus free points are included in the validation dataset.

### 4.2.1 Cirrus classification

The CCF of CiPS and COCS and the OPF of CiPS are evaluated as a function of the geographic position. This aspect is interesting due to the very different meteorological conditions present on the SEVIRI disc. Figure 6a and 6b shows the gridded FAR (Eq. (4)) for the CCF of CiPS and COCS respectively over 5°×5° boxes, using the V3 CALIOP products as reference.

5    As mentioned in Sect. 3.6 the average FAR for the CiPS cirrus detection is 3.9 %. The FAR is sensitive to the frequency of the events, meaning that over regions where the natural probability of cirrus presence is high, a single false alarm will have a larger impact on the total FAR than over regions where the natural probability of cirrus presence is low. Although the FAR of CiPS is relatively homogeneous across the SEVIRI disc, this effect can be observed with higher FARs along the ITCZ and lower FARs over the Sahara desert for example.

COCS has an equally low FAR over arid regions, but has a clearly higher FAR in general. In particular over icy surfaces like Greenland and Antarctica, COCS overestimates the cirrus presence, with FARs up to approx. 90 %. But for high latitudes in general, the FAR of COCS remains higher than CiPS. In the polar regions (latitude $\geq 65°$ N/S) the average FAR is 33 % for COCS and 5.3 % for CiPS. Also over Europe the FAR of CiPS is clearly lower. Furthermore, COCS strongly overestimates the cirrus presence around the sub-satellite point of SEVIRI. For viewing zenith angles smaller than 15°, COCS has an average FAR of 23 %. This deficiency is not shown by CiPS, which has an average FAR of 8.5 % for the same area. Furthermore, a false alarm of COCS has $IOT_{COCS} \geq 0.1$, whereas a false alarm of CiPS can have an $IOT_{CiPS}$ down to 0.0, i.e. $IOT_{CiPS} > 0.0$.

Due to the high probability of cirrus cloud presence along the ITCZ, the effect of the higher FAR of CiPS over this region is small, since a high cirrus probability prevents false alarms from occurring. Figure 6c and 6d show the total number of false alarms/positives $N_{FP}$ by CiPS and COCS respectively, i.e. the total number of cirrus free points in the validation dataset (approx. 3.3 millions) that are falsely classified as cirrus. Again the numbers are calculated over 5°×5° boxes. Even if the probability of having a false alarm by CiPS is higher than the average FAR along the ITCZ (Fig. 6a), the absolute number of false alarms is just as high as for most regions across the SEVIRI disc (Fig. 6c). Looking at $N_{FP}$ by COCS (Fig. 6d), more false alarms are observed at high latitudes (especially over icy surfaces), over Europe and around the sub-satellite point.

The FAR can easily be optimised by reducing the number of detected cirrus clouds (see Fig. 3). Thus it is necessary to simultaneously look at the performance in cirrus detection alongside the false alarm analysis. A reduced POD (probability of detection) would be a natural effect if the FAR is reduced, but despite the low FAR of CiPS the POD remains high. Fig. 7 shows the POD of CiPS, again in comparison to COCS. The POD is a function of $IOT_{CALIOP}$ and within each $IOT_{CALIOP}$ interval the POD given by Eq. (3) is calculated, using the V3 CALIOP products as reference. For a better visualisation the POD is presented with a logarithmic scale for $IOT_{CALIOP} < 1.0$ and with a linear scale for $IOT_{CALIOP} \geq 1.0$. For cirrus clouds with $IOT_{CALIOP} > 1.0$, CiPS and COCS perform similarly. A strong difference is instead seen for the thin cirrus clouds, where CiPS detects more cirrus clouds compared to COCS. For example at $IOT_{CALIOP} = 0.1$, CiPS detects 71 % of the cirrus clouds and COCS 43 %. A higher POD for thin cirrus clouds is an important improvement when studying contrail cirrus or the cirrus life cycle for example. Figure 7 does only present the results for the transparent cirrus clouds where the CALIOP laser was not saturated. For the opaque cirrus clouds the average POD is 98 % for both CiPS and COCS. The geographical dependency of POD is clearly anti-correlated with the geographical dependency of the FAR, meaning that CiPS has its highest and lowest POD over regions where the natural probability of cirrus presence is high and low respectively. Apart from that, the POD of CiPS is homogeneous across the SEVIRI disc.

Fig. 8 shows the FAR of the CiPS OPF, again over 5°×5° boxes, using the V3 CALIOP products as reference. Since the OPF is a new variable introduced with CiPS, the results cannot be compared to COCS. As mentioned in Sect. 3.6 the average POD and FAR is 71 % and 4.0 % respectively. Both quantities are relatively homogeneous across the SEVIRI disc, but the risk of falsely classifying a transparent cirrus cloud as opaque is slightly lower in the tropical regions (latitude < 30° N/S).

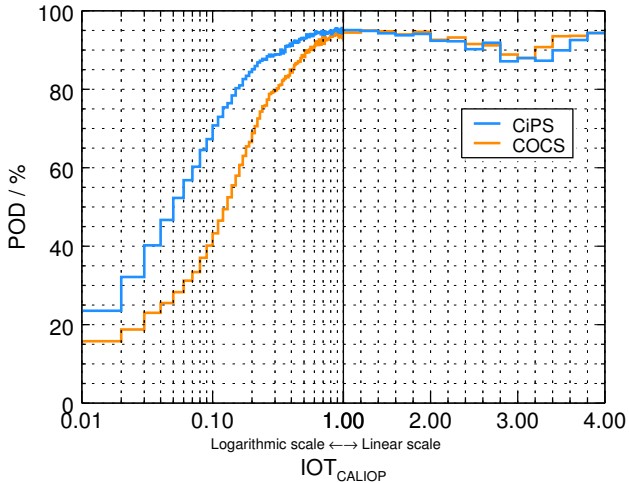

**Figure 7.** The POD of CiPS and COCS as a function of the IOT retrieved by CALIOP.

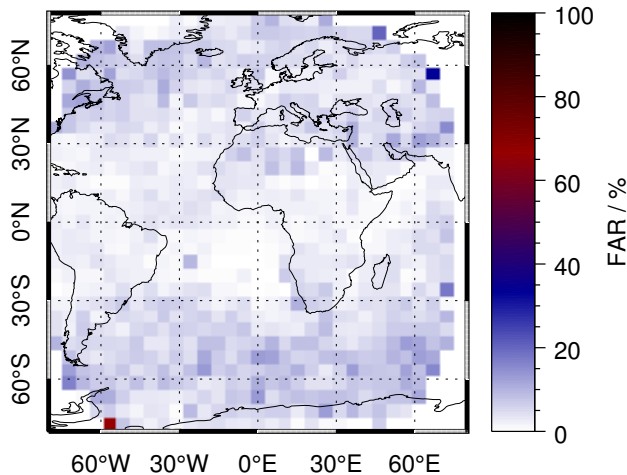

**Figure 8.** FAR of the CiPS OPF (opacity flag).

### 4.2.2 Cirrus properties

Figure 9 shows two density scatter plots, with $CTH_{CALIOP}$ on the horizontal axes and $CTH_{CiPS}$ (left) and $CTH_{COCS}$ (right) on the vertical axes. The color shows the normalised relative frequency, which is the relative frequency normalised to the interval 0–1. Along with the scatter plots the MPE and MAPE (Eq. (5) and 6) of CiPS and COCS with respect to CALIOP as a function

5   of $CTH_{CALIOP}$ is shown (right). CiPS and COCS are validated using their own respective cirrus flags, meaning that the $CTH_{CiPS}$ is validated using the cirrus covered points that CiPS detects, whereas the $CTH_{COCS}$ is validated using those cirrus covered points that COCS detects. Using a common cirrus flag (i.e. those cirrus covered points that both CiPS and COCS detect) shows

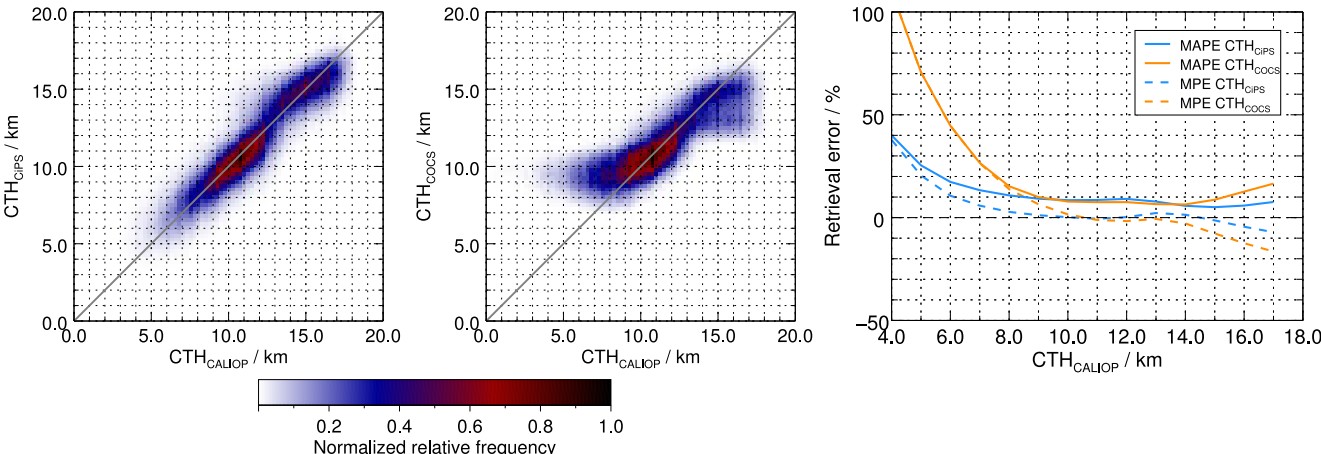

**Figure 9.** Density scatter plots with the CTH retrieved by (left) CiPS and (centre) COCS on the vertical axes and the corresponding V3 CALIOP data on the horizontal axes. The grey lines represent the 1-1 line. (Right) The MAPE (solid) and MPE (dash) of the CTH retrieved by CiPS and COCS with respect to the CTH measured by CALIOP.

marginal differences, with slightly reduced errors for CiPS, as a result of the reduced amount of very thin cirrus that only CiPS detect, for which the CTH is more difficult to accurately estimate.

With CiPS the CTH is retrieved with a higher accuracy compared to COCS, especially for high and low cirrus clouds. The correlation between CALIOP and CiPS is 0.90. For CALIOP and COCS, the correlation coefficient is 0.82.

The MPE shows that CiPS overestimates and underestimates the CTH of the lowest and highest cirrus clouds respectively, even if the errors are smaller than for COCS. From 8-15 km the MPE is close to zero, meaning that the CTH retrieval by CiPS is unbiased in this $CTH_{CALIOP}$ range. The MAPE shows that the average magnitude of the CiPS error is 10 % or less for cirrus clouds having a CTH above 8 km. Furthermore, the MAPE clearly shows the better accuracy of CiPS. For example for cirrus clouds with a $CTH_{CALIOP}$ between 4–5 km, representing mid-level clouds with icy tops, the MAPE is 38 % for CiPS. For

COCS the corresponding number is 107 % with solely overestimated values (MAPE = MPE). This is mainly an effect of the CTH filtering used for COCS (Sect. 2.5), which excluded cirrus clouds with a $CTH_{CALIOP} < 4.5$ km from the training dataset, leading to strong overestimations of lower values. Furthermore, this type of low cirrus/icy clouds are found in the polar regions (see Fig. 10b), where the retrieval conditions for SEVIRI are more challenging with larger viewing zenith angles and pixel sizes.

The CTH has a strong latitude dependency and the CiPS results shown in Fig. 9 are not representative for all latitudes. Figure 10a shows the MPE of the $CTH_{CiPS}$ retrievals with respect to CALIOP as a function of $CTH_{CALIOP}$ and the latitude. Figure 10b shows the corresponding occurrences of the points that make up the statistics shown in Fig. 10a. Please remember that the validation dataset is a random subset of CALIOP data collected over a time period of almost six years and hence represents the natural latitudinal distribution of cloud top heights.

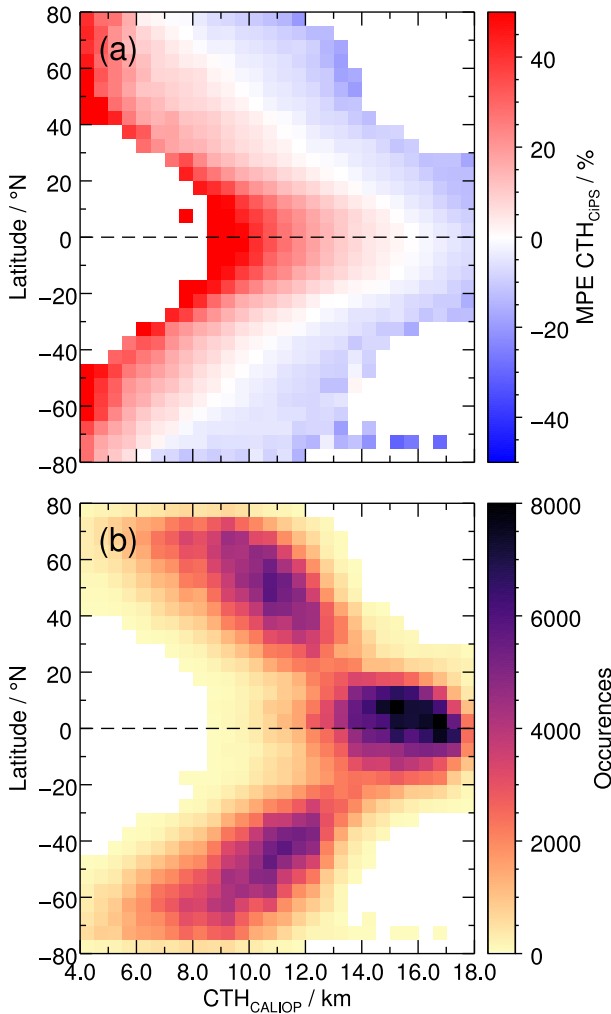

**Figure 10.** (a) 2D histogram showing the MPE of the $CTH_{CiPS}$ retrieval as a function of the reference CTH retrieval by CALIOP and the latitude. (b) The corresponding occurrences of the points that make up the statistics shown in (a).

The MPE shows a clear latitude dependency and in contrast to Fig. 9c, where CiPS is shown to have no bias (MPE $\approx 0$) between 8–15 km, we see that the $CTH_{CALIOP}$ limit when CiPS starts to over- and underestimate the CTH increases towards the equator. At higher latitudes (e.g. over Europe), we see that CiPS is more likely to underestimate the CTH also for lower $CTH_{CALIOP}$ around 11-14 km, with an increasing bias towards higher latitudes. Similarly the $CTH_{CiPS}$ for cirrus clouds with
5  $CTH_{CALIOP} < 13$ km is more likely to be overestimated along the ITCZ, with increasing errors towards the equator. From Fig. 10b it is clear that the situations with higher errors and stronger biases ($|MPE| \gtrsim 20\,\%$) are comparably rare and that $CTH_{CiPS}$ is unbiased for the more frequent combinations of $CTH_{CiPS}$ and latitude.

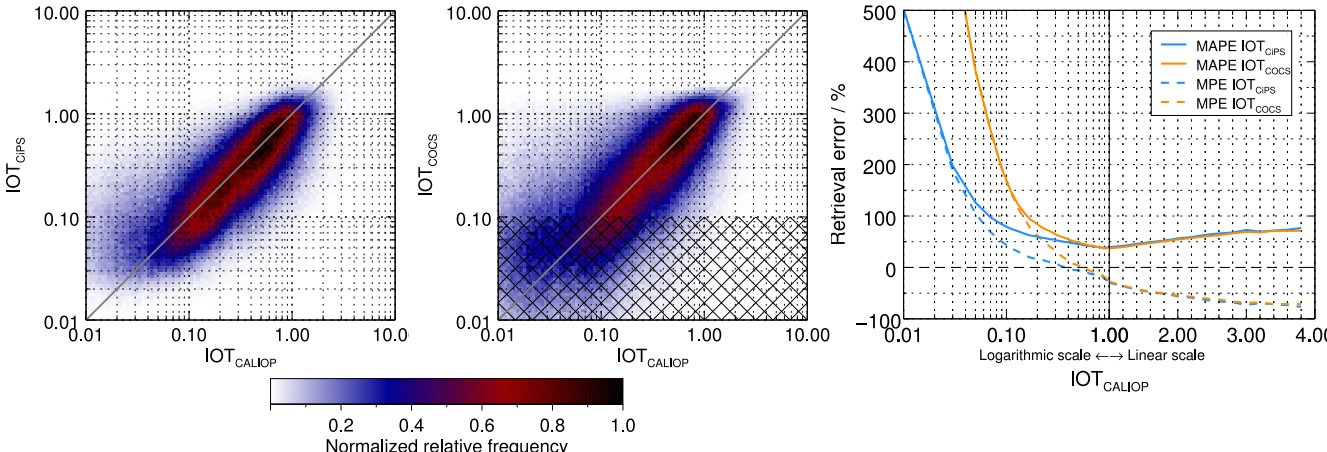

**Figure 11.** Density scatter plots with the IOT retrieved by (left) CiPS and (centre) COCS on the vertical axes and the corresponding V3 CALIOP data on the horizontal axes. The grey lines represent the 1-1 line. (Right) The MAPE (solid) and MPE (dash) of the IOT retrieved by CiPS and COCS with respect to the IOT retrieved by CALIOP. The black grid on top of the right scatter plot illustrates the area where COCS does not detect any cirrus clouds as a results of the COCS cirrus detection threshold at $IOT_{COCS} = 0.1$ (Sect. 2.5).

Remark the difference between the CiPS CTH retrieval and standard ones (e.g. Menzel et al., 2008), where the determination of cloud top height requires the knowledge of the appropriate vertical temperature profile from NWP (numerical weather prediction) models, while CiPS only requires the surface skin temperature from a NWP along with the SEVIRI brightness temperatures and auxiliary data.

Figure 11 shows again two density scatter plots, now with $IOT_{CALIOP}$ on the horizontal axes and $IOT_{CiPS}$ (left) and $IOT_{COCS}$ (centre) on the vertical axes. As before the color shows the normalised relative frequency. Again the MPE and MAPE (Eq. (5) and (6)) of CiPS and COCS with respect to CALIOP as a function of $IOT_{CALIOP}$ is shown in the right panel. Only transparent cirrus clouds, where CALIOP was not saturated, are included here. The two algorithms are validated using their respective cirrus cloud flags (as explained above for the CTH). This is not 100 % true for the $IOT_{COCS}$ scatter plot however, where all

points with a retrieved $IOT_{COCS} > 0.0$ are included. Instead the black grid on top of the scatter plot illustrates the area where COCS does not detect any cirrus clouds as a results of the COCS cirrus detection threshold at $IOT_{COCS} = 0.1$ (Sect. 2.5). A relatively large scatter is observed for both algorithms. CiPS shows a better correlation with the CALIOP retrievals though. The correlation between CiPS and CALIOP is 0.65, whereas the correlation between COCS and CALIOP is 0.61. Furthermore, CiPS shows higher frequencies along the 1-1 line down to $IOT_{CALIOP} \approx 0.09$, but also below this value the correlation between

CALIOP and CiPS is evident. Only below $IOT_{CALIOP} = 0.04$ the correlation gets lost.

For a better visualisation of the lower IOT range, where most points are located, the density scatter plots have logarithmic axes. This does however visually reduce the errors, so for a quantitative evaluation attention should be paid to the right panel in Fig. 11 showing the MPE and MAPE of CiPS and COCS with respect to CALIOP. The MPE and MAPE are functions of $IOT_{CALIOP}$ and again the results are presented using a logarithmic scale for $IOT_{CALIOP} < 1.0$ and a linear scale for $IOT_{CALIOP} \geq$

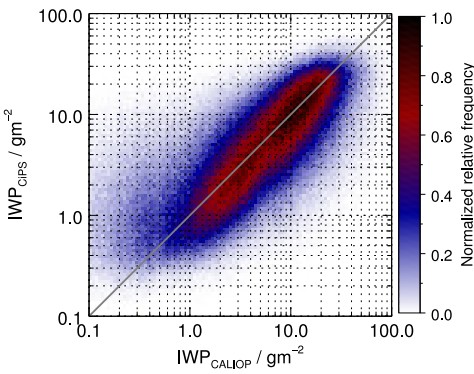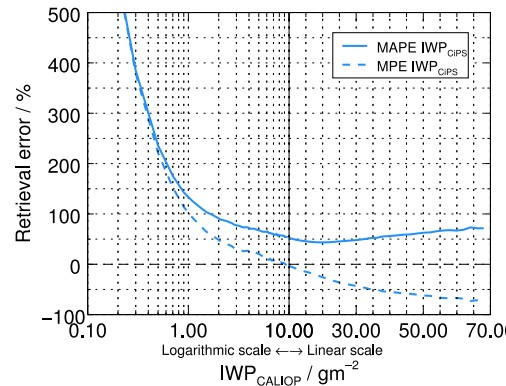

**Figure 12.** (Left) Density scatter plot with the IWP retrieved by CiPS on the vertical axis and the corresponding V3 CALIOP data on the horizontal axis. The grey line represents the 1-1 line. (Right) The MAPE (solid) and MPE (dash) of the IWP retrieved by CiPS with respect to the IWP retrieved by CALIOP.

1.0. From the MAPE the low accuracy of CiPS for sub-visual cirrus clouds becomes evident. For $IOT_{CALIOP} < 0.03$, we also see that MAPE = MPE, meaning that CiPS entirely overestimates the IOT in this region. For COCS, the same is observed for $IOT_{CALIOP} < 0.1$ as a direct effect of the inability of COCS to detect cirrus clouds with an $IOT_{COCS} < 0.1$. The opposite is observed for thicker cirrus clouds ($IOT_{CALIOP} \gtrsim 2.0$), where both CiPS and COCS entirely underestimate the IOT (MAPE =

$-$MPE). With CiPS the IOT can be retrieved with a MAPE of 50 % or less for cirrus clouds with $0.35 \lesssim IOT_{CALIOP} \lesssim 1.8$. Similarly the MAPE of the retrieved $IOT_{CiPS}$ is 100 % or less for cirrus clouds with $IOT_{CALIOP} > 0.07$ and 230 % or less down to sub-visual cirrus clouds ($IOT_{CALIOP} > 0.03$). The corresponding MAPE for the IOT retrieved by COCS within the same $IOT_{CALIOP}$ intervals are 59, 290 and 720 %. A MAPE of 100 % might seem high, but one should keep in mind that this translates into small absolute errors for such thin cirrus clouds. For the lower $IOT_{CALIOP}$ range, a similar scatter is observed

between $IOT_{CALIOP}$ and modelled IOT from IR radiances for thin cirrus clouds in Holz et al. (2016).

Figure 12 shows the density scatter plot with $IWP_{CALIOP}$ on the horizontal axis and $IWP_{CiPS}$ on the vertical axis (left) together with the MPE and MAPE (Eq. (5) and (6)) of CiPS with respect to CALIOP as a function of $IWP_{CALIOP}$ (right). Please note that again the density scatter plots have logarithmic axes and the errors are presented using logarithmic scale for the thinner cirrus clouds ($IWP_{CALIOP} < 10.0$) and with linear scale for the thicker ones ($IWP_{CALIOP} \geq 10.0$). Since the IWP is not retrieved

by COCS, no additional results are shown here for comparison. Again only transparent cirrus clouds are included.

The scatter between $IWP_{CiPS}$ and $IWP_{CALIOP}$ is very similar to the one between $IOT_{CiPS}$ and $IOT_{CALIOP}$. This is not surprising since the IWC from CALIOP is retrieved from the measured extinction coefficients using a parametrisation. The correlation between CiPS and CALIOP is however slightly lower for the IWP retrieval (0.59) compared to the IOT retrieval. This is also expected since possible deficiencies in the CALIOP IWC parametrisation will make it more difficult for the ANN to learn

the relationship between the input data and the IWP. Nevertheless, these results show that the ANN is capable of reproducing this relationship in a good way. With CiPS the IWP can be retrieved with a MAPE of 100 % or less for cirrus clouds with

$IWP_{CALIOP} > 1.7\,gm^{-2}$ and 200 % or less down to $IWP_{CALIOP} \approx 0.7\,gm^{-2}$. Please notice that deviations of 100 % are common even when microwave information is considered (e.g. Holl et al., 2014, , even if their error measure is different from ours).

In contrast to the $CTH_{CALIOP}$ retrieval, CiPS shows a stable performance for the IOT and IWP retrievals across all latitudes (not shown here). The only anomaly observed is that the CiPS retrieval errors for thin to sub-visual cirrus are lower over
convergence zones like the ITCZ, where they are mostly found (Sassen et al., 2009; Martins et al., 2011).

As expected and as seen in Fig. 9, 11 and 12, CiPS is not able to perfectly model the CALIOP cirrus properties using the SEVIRI, ECMWF and auxiliary data. There are several sources of error that add to the final performance of CiPS. Most importantly CALIOP and SEVIRI have different sensitivities to cirrus clouds. This is especially clear for thin to sub-visual cirrus clouds where CALIOP is able to accurately retrieve the top height and optical properties. Such faint cirrus leave a
considerably weaker or no mark on the SEVIRI observations though, making it difficult to inversely determine the cirrus properties. Similarly the CTH is not necessarily defined equally by CALIOP and SEVIRI, as CALIOP is able to discern thinner icy layers at the cloud top, that may appear as "invisible" to SEVIRI. Also for thicker cirrus clouds where both CALIOP and SEVIRI (thermal observations) approaches the point of saturation, the different sensitivities lead to ambiguous collocations. When an ANN is trained with a set of different output values that correspond to approximately the same input data as a
result of the lower sensitivity, the ANN will not be able to model an accurate relationship. The reason for this is that the input vector contains no information on how the difference in sensitivity affects the target values. This can be regarded as an unknown hidden variable. This is not an ANN specific weakness, but applies to all regression models minimising the squared error. When such a set of incomplete input data (in the sense that there is a strong hidden variable) is given to the final ANN, it will output a conservative mean value that can be understood as an average over the distribution of the most likely solutions weighted
by their probability. The larger the difference in sensitivity the higher will the variance within the distribution of the most likely solutions be, leading to larger retrieval errors. Throughout most of the output data range this error will be random. But obviously, the distribution of the most likely solutions cannot be centred around the extreme values leading to systematic over- and underestimations of low and high output values when a conservative mean value is calculated. This effect increases towards the extreme values as the desired output value is skewed towards the edge of the distribution of the most likely solutions. This
effect is clearly seen in Fig. 11c and 12c where low and high $IOT_{CALIOP}/IWP_{CALIOP}$ are over- and underestimated respectively. This is to some extent also seen for the $CTH_{CiPS}$ retrieval in Fig. 9c, especially for low $CTH_{CALIOP}$. Due to the randomness of the effects a lower sensitivity introduces, adding information about the magnitude of the sensitivity to the input vector is not likely to improve this situation. The larger $CTH_{CiPS}$ retrieval errors observed for low clouds can also be attributed to the smaller temperature contrast with respect to the surface temperature and thus the weaker radiative signal that those clouds have
compared to higher cirrus clouds. Another source of error that amplifies the effect discussed above, is the risk that there are additional variables relevant for finding an accurate relationship that are not represented by the vector of input data.

As discussed in Sect. 3.4.1 imperfect collocations as a result of the different spatial scales of CALIOP and SEVIRI together with partial cloud cover or spatially inhomogeneous clouds will further add to the retrieval errors. In a situation where CALIOP observed a small optically thin area of an otherwise optically thick cirrus inside a SEVIRI pixel, CiPS is likely to overestimate

IOT$_{CALIOP}$ and IWP$_{CALIOP}$. Similarly if CALIOP observed a small optically thick area of an otherwise optically thin cirrus inside a SEVIRI pixel, CiPS is likely to underestimate IOT$_{CALIOP}$ and IWP$_{CALIOP}$.

## 5   The cirrus life cycle with CiPS

In this section the potential of CiPS is illustrated by analysing the temporal evolution of a thin cirrus cloud throughout its life cycle. The life cycle of natural cirrus and contrails is an important aspect to study (Szantai et al., 2001; Luo and Rossow, 2004; Vazquez-Navarro et al., 2015), since knowledge about the physical processes that govern their life cycle is essential for an accurate representation in weather and climate models.

Here we analyse the life cycle of an outflowing cirrus originating from an orographic cirrus. The cirrus cloud was identified south of the Pyrenees on September 26.09.2014 at 10.00 UTC from SEVIRI. A false color RGB for this scene including the contour of the CiPS cirrus mask is shown in Fig. 13a. Using the binary cirrus cloud masks obtained with CiPS and 2D image correlation the detected cirrus cloud is tracked backward and forward in time using the rapid scanning service of SEVIRI with a temporal resolution of 5 min. A similar method is used to track cloud patterns in **?**. The minimum bounding box enclosing the selected cirrus cloud is cross-correlated with the previous/next cirrus cloud mask in order to find the position of the cirrus cloud 5 min earlier/later. The scene having the highest correlation with this bounding box is identified. A cirrus cloud patch within this scene is considered part of the tracked cirrus if it is completely or partly covered by the tracked cirrus from the previous scene. This allows for a simultaneous tracking of multiple cirrus clouds in the likely event of the tracked cirrus cloud breaking up into multiple smaller cloud patches (Fig. 13b). All cirrus clouds smaller than 5 SEVIRI pixels are filtered out. Using the CiPS opacity flag, it was concluded that the tracked cirrus cloud was transparent throughout the life cycle, indicating that the true, rather than apparent, IOT and IWP can be derived by CiPS.

The path and temporal evolution of the cirrus cloud with a temporal resolution of 120 min (2 h, apart from the first and the last step) is visualised in Fig 13b. The starting time is 5.25 UTC on 26.09.2014, while the plot ends at 0.55 UTC on 27.09.2014. Notice that the time axis runs from the right to the left in order to follow the cirrus cloud that moves from the East to the West. We see that the cirrus cloud formed from several small cirrus patches originating from the outflow of the orographic cirrus south of the Alps and moved westwards over the Mediterranean Sea and Spain before it attached to another larger cirrus cloud over the Atlantic Ocean. By tracking multiple cloud patches simultaneously the cirrus cloud can be monitored as a whole, even when it splits into several parts. Throughout the life cycle, a maximum number of 24 cirrus cloud patches were tracked and analysed simultaneously as one cirrus cloud. Triggering the tracking 2 h before/after the starting time presented here (10:00 UTC) results in only marginal differences ($< 5.0\,\%$ in horizontal area, not shown here), as some small cirrus patches that in the end form the tracked cirrus might be temporarily missed. This validates the robustness of the tracking method.

To validate the tracking method, the tracking was initialised at 08:00 UTC and 12:00 UTC as well, with onl

The temporal evolution of the cloud horizontal area can be seen at full temporal resolution (5 min) in Fig. 14a. The same figure also presents the temporal evolution of the CTH, IOT and IWP retrieved by CiPS.

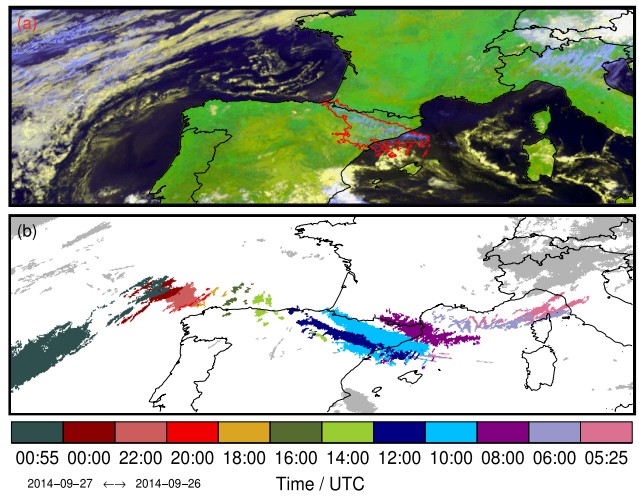

**Figure 13.** (a) False color RGB composite on September 26, 2014 at 10.00 UTC. The red contour of the CiPS cirrus cloud mask shows the outline of the cirrus cloud, whose life cycle is analysed. The orographic cirrus, from which the tracked cirrus originates, is clearly seen south of the Alps. (b) The path and temporal evolution of the cirrus cloud as it is tracked backward and forward in time with a temporal resolution of 120 min. The light grey color shows all cirrus clouds present at 05.25 UTC that were not tracked in order to understand the origin of the analysed cirrus cloud.

The cirrus cloud detaches from the orographic cirrus at 05.25 UTC and starts to grow in size immediately. The IOT and IWP decrease for the first 30 min, but start to grow along with the horizontal area at around 06.00 UTC. The lower IOT and IWP quartiles grow comparably slow and the increased mean values are a result of an increased fraction of thicker pixels, which is indicated by steeper curves of the medians and upper quartiles. The cirrus grows in size, IOT and IWP for 4 h, before it
reaches its maximum horizontal area of nearly 60 000 km$^2$ at around 10.00 UTC. During this time period the CTH increases slightly, but remains comparably stable, i.e. the effect of the Pyrenees, that are reached by the cloud at ca. 07.00 UTC, on CTH is small. At around 09.15 UTC the cloud starts to sink and ca. 1 h later the cloud starts to decrease in size, indicating that sufficiently warm temperatures have been reached, forcing the cloud to dissipate. Despite the dissipation, the average IOT and IWP continue to grow for another hour, reaching an average IOT$_\text{CiPS}$ and IWP$_\text{CiPS}$ of 0.23 and 4.2 gm$^{-2}$. This is observed
because the comparably large areas of thin cirrus with low IOTs and IWPs are the first to dissipate, leading to smaller fraction of low IOTs and IWPs and thus higher mean values. This is confirmed by the lower quartiles that start to increase more strongly when the horizontal area turns downward at around 10.30 UTC.

The IOT and IWP start to decrease at around 11.30 UTC and continue to do so until 19.00 UTC, when just a few small and thin cirrus cloud patches remain with average IOT$_\text{CiPS}$ of 0.07 and average IWP$_\text{CiPS}$ of 1.0 gm$^{-2}$. For the same period we see
that the cloud slowly starts to gain altitude and around 19.00 UTC the altitude is high enough for the cloud to once again start to grow in size, IOT and IWP. The IOT and IWP grows marginally, again as a results of an increasing fraction of thicker pixels (stable lower quartiles). The growth in size is more evident and the horizontal area increases from 2800 to 19 200 km$^2$ during

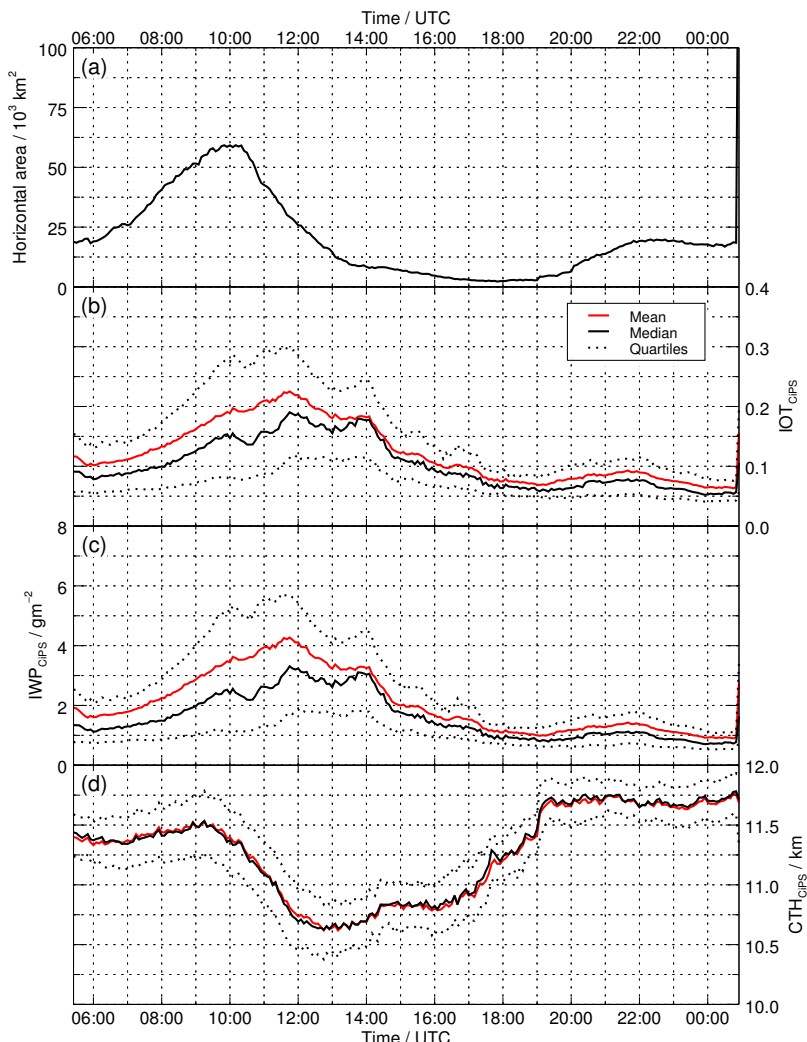

**Figure 14.** Temporal evolution of the cloud properties for the cirrus described in Fig. 13 with a temporal resolution of 5 min: (a) the horizontal cirrus cloud area, (b) IOT, (c) IWP and (d) CTH. For the IOT, IWP and CTH the mean, median, upper and lower quartile values are presented.

the 3 h long period of growth. Finally the horizontal area, IOT and IWP are slightly reduced before the tracked cirrus cloud connects to another cirrus cloud at 00.55 UTC. This is seen in Fig. 13b and by the rapid growth in size, IOT and IWP. The CTH remains constant, which tells us that the other cirrus cloud in fact is located at a similar altitude.

## 6 Conclusions

The CiPS algorithm presented in this work detects cirrus clouds and retrieves their CTH (cloud top height), IOT (ice optical thickness) and IWP (ice water path) along with an OPF (opacity flag) using SEVIRI, ECMWF and auxiliary data. CiPS utilises a set of four artificial neural networks, trained with V3 CALIOP L2 layer data as a reference. CiPS does not take advantage of the SEVIRI channels with significant solar contribution and can thus be used during both day and night. By using ANNs, the idea is to combine the high sensitivity and vertical resolution of CALIOP with the large spatial coverage and high temporal resolution of SEVIRI. Thus, the ultimate goal of CiPS is to retrieve CALIOP-like cirrus properties for the full SEVIRI disc (approx. one third of the Earth) every 15 min.

CiPS shows a good performance when validated against independent CALIOP data. CiPS detects 95 % of all cirrus clouds with an optical thickness of 1.0 and 71 % of all cirrus clouds with an optical thickness of 0.1. On average, CiPS correctly classifies 96 % of the cirrus free pixels. For cirrus clouds with $0.35 \lesssim IOT_{CALIOP} \lesssim 1.7$, the IOT can be retrieved with a MAPE of 50 % or less, relative to CALIOP. For cirrus clouds with $IOT_{CALIOP} \gtrsim 0.07$, CiPS retrieves the IOT with a MAPE of 100 % or less. For thinner clouds, where the cirrus signal in the SEVIRI channels is weak, the error increases, but is still 230 % or less for $IOT_{CALIOP} \gtrsim 0.03$ (sub-visual cirrus). The IWP retrieved by CiPS has a similar performance, but a larger MAPE for the thinner clouds. This is expected since the IWP is parametrised from the CALIOP extinction coefficients, which means that deficiencies in the parametrisation will make it more difficult for CiPS to learn the relationship between the input and output variables during training. The CTH, which is directly measured by CALIOP, is also the variable that CiPS retrieves with the highest accuracy. For cirrus clouds with $CTH_{CALIOP} \geq 8$ km, the MAPE is 10 % or lower. Since CALIOP is unable to penetrate thicker cirrus clouds, an additional ANN is trained to determine whether a cirrus cloud is opaque or not (as seen from CALIOP). 96 % of the transparent cirrus clouds that CiPS detects are correctly classified as transparent. Similarly, 71 % of the opaque cirrus clouds that CiPS detects are correctly classified as being opaque. This information is very important to discern thin cirrus, for which CiPS works very well, from thicker clouds where neither CiPS nor CALIOP can capture the complete IOT and IWP. The reported errors of CiPS are only with respect to CALIOP. Additionally CiPS, as an ANN, will have inherited any error that the CALIOP products have with respect to the true cirrus properties.

CiPS has a better performance in all aspects with respect to COCS, another algorithm that uses ANNs for retrieving the CTH and IOT from SEVIRI using CALIOP as reference (Kox et al., 2014). Significant improvements have been made for the detection of the thinner cirrus clouds and the retrieval of the corresponding IOT. Also for the higher and lower cirrus clouds, the CTH retrieval has been clearly improved. Furthermore, IWP and an OPF have been added. Improvements with respects to COCS can be attributed to several factors. 1) We use new input data including the modelled surface skin temperature and the regional maximum and average brightness temperatures. 2) The training meta-parameters and ANN structures have been thoroughly investigated and optimised for CiPS. 3) The training of CiPS was more rigorous, with mini-batch learning rather than stochasitc learning as well as a tuning phase with gradually increasing batch size and gradually decreasing learning rate and momentum. Furthermore an internal validation dataset was used during the training of CiPS in order to monitor the accuracy and avoid overfitting. 4) The use of the more accurate V3 CALIOP data allowed us to omit the CTH filtering used

for COCS, leading to a more accurate CTH retrieval by CiPS. 5) CiPS utilises multiple ANNs. COCS uses one single ANN trained with cirrus covered as well as cirrus free pixels. On the contrary, the CiPS ANNs that retrieve the CTH, IOT, IWP and OPF were trained exclusively with cirrus covered pixels, resulting in lower retrieval errors of CiPS. The larger retrieval errors of COCS for thin cirrus clouds also affects the IOT dependent cirrus cloud detection of COCS, with both a lower POD and a higher FAR compared to CiPS.

As an application example the life cycle of a thin cirrus cloud and the temporal evolution of its properties is investigated. The cirrus cloud lives for nearly 20 h and is shown to originate from outflowing cirrus cloud patches from an orographic cirrus cloud. By analysing the cirrus properties retrieved by CiPS, the physical processes throughout the cirrus life cycle can be better understood.

The approach of using ANNs is very fast and requires little computational power compared to standard physical methods that require extensive radiative transfer calculations and/or interpolation in a multi-dimensional space. On a common standard PC, one complete SEVIRI image with $3712 \times 3712$ pixels is processed in approx. 60 s including the cirrus detection and the CTH, IOT, IWP and OPF retrieval. By training multiple ANNs with different numbers of hidden layers and hidden neurons, we see that a larger network with more hidden layers and hidden neurons does generally provide a higher POD and lower errors. A larger network does however come at the expense of more computational power, especially for the training but also for the application.

With CiPS we are now able to study the temporal evolution, life cycles and diurnal cycles of thin cirrus clouds, natural and anthropogenic (contrails), including their coverage, CTH, IOT and IWP with a higher degree of accuracy. The inclusion of a physical variable like the IWP further allows for direct comparison with weather, climate or large eddy simulation models.

As a next step, the CiPS retrievals will be further characterised with respect to the underlying surface type and the presence of aerosol layers and liquid water clouds below the cirrus. Constant developments and improvements of the CALIOP cirrus cloud retrievals also opens the door for further improvements of CiPS. Another aspect of improvement would be to introduce new input data, for example temperature and humidity profiles and surface emissivity. Although this manuscript is limited to CALIOP retrievals, one could investigate the usefulness of synergistic CALIOP/CloudSat retrievals as training reference data. One could also investigate the usefulness of a more rigorous balancing of the training dataset in order to reduce the number of training points without loosing any unique information.

## Appendix A:  List of abbreviations

ANN      Artificial Neural Network

CCF      Cirrus Cloud Flag

CTH      Cloud Top Height

FAR      False Alarm Rate

IOT      Ice Optical Thickness

ITCZ     Inter-Tropical Convergence Zone

IWP      Ice Water Path

MAPE    Mean Absolute Percentage Error

MPE     Mean Percentage Error

MLP     Multi-Layer perceptron

OPF     Opacity Flag

POD     Probability of Detection

*Acknowledgements.* This research was supported by the DLR (Deutsches Zentrum für Luft- und Raumfahrt)/DAAD (Deutscher Akademischer Austauschdienst) Research Fellowship Programme für Doktoranden, 14.

We thank the NASA Atmospheric Science Data Center for their kind support and for providing the V3 CALIOP layer data in a subsetted form. We also thank Mark Vaughan for his guidance on how to properly account for the vertical overlap of cloud and aerosol features in the CALIOP layer products. We want to express our gratitude to Diego Loyola for an interesting and helpful discussion about the application of ANNs in satellite remote sensing. We also thank Stephan Kox for the discussion on COCS and the relevant routines that that were provided. We gratefully acknowledge the constructive comments of two anonymous reviewers, Florian Ewald, André Butz and Ulrich Schumann, that greatly improved the quality and clarity of this manuscript.

The SEVIRI data were provided by EUMETSAT (European Organisation for the Exploitation of Meteorological Satellites) and the modelled surface temperature was obtained from ECMWF (European Centre For Medium-Range Weather Forecasts).

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
