# Peer review of "Cirrus cloud retrieval with MSG/SEVIRI using artificial neural networks"

_Atmospheric Measurement Techniques, 2017_

## Referee Comment (RC1) · Anonymous Referee #1 · 24 Apr 2017

The authors present a new retrieval method retrieving three cirrus-related climate variables from SEVIRI. The science appears solid and is mostly well-described. I recommend publication with minor revisions. Below I make a few general comments, followed by detailed comments line by line, which are mostly requests for clarification.

**General comments**

The authors characterise the performance of CiPS in several ways. Did you look at its performance depending on underlying surface type? Snow surfaces are famously difficult, but other surface aspects may be relevant as well.

The elephant in the room in many retrieval products, in particular those based on machine learning, is the uncertainty. Although the the paper provides characteristics on overall performance, is there any way to get an uncertainty estimate for a specific retrieval?

Is there, or is there planned to be, a publicly available data product based on CiPS, so that people can download the data and explore it on their own? I think there should be people interested in using it.

Could the approach be extended to other imagers than SEVIRI, as long as those have an overlap with CALIOP to be trained with? Or are the properties of SEVIRI (footprint size? scan speed? channels?) essential for CiPS to work?

**Specific comments**

Abstract

- Page 1, lines 1–2: replace "one of the largest uncertainties" by "one of the largest sources of uncertainty", and replace "they" by "their physical properties"

- Page 1, line 8: after 71, add %

1. Introduction

- Page 3, line 29: add "piece of" before "information"

- Page 3, line 32: please explain acronym LES (I guess this is large eddy simulation in the context of a cloud resolving model), which is missing in the text and in Appendix A. As this acronym appears to be used only once in the paper, I suggest just writing it out and avoiding the acronym altogether.

2.1. SEVIRI

Please expand this paragraph with:

- longitude above which SEVIRI is located (finally described on page 12, line 22)

- total range of field of view of the disk SEVIRI can observe

2.2 CALIOP

- Page 4, line 18: This usage of the word "frequency" is potentially confusing, maybe write that it measures 20.16 times per second or every 49.6 ms (when I see the word frequency I think of electromagnetic frequency).

- Page 4, line 24: please explain acronym IOT

- Page 4, line 26: This line has some typesetting issues: CPL should be explained at first use, and the formatting of the citation is incorrect.

2.3.2 Learning through backpropagation

- Page 6, line 16: replace "a" by "an".

2.4 Validation metrics

- Page 8, equation 5: this MPE metric is risky because over- and underestimates can cancel out each other. I realise that is why the authors also look at MAPE but I think this risk should be explicitly pointed out.

3 CiPS

- Page 9, line 4: remove "though"

- Page 9, line 5: remove "the"

3.1 Multiple artificial neural networks

- Page 9, line 10: Remove "decimal" before "number". I don't think the authors actually mean a decimal number as defined by IEEE 754-2008, presumably it's a regular binary in their software implementation.

- Page 9, line 23-24: Does CALIOP (reliably) identify when it is saturated?

- Page 10, line 2: The authors refer to "photon counts" but I don't expect SEVIRI actually counts photons. The digital count level is probably rather a conversion from a voltage. I assume the authors use brightness temperatures already calibrated elsewhere, so I suggest to cite the relevant paper or technical report if available.

3.2.1 Brightness temperatures from SEVIRI

- Page 10, line 8: It would be useful to remind the reader to what surface area $19 \times 19$ pixels$^2$ corresponds ($57 \times 57$ km$^2$?)

3.3 Output data: cirrus properties from CALIOP

- Page 12, line 3: Which spatial resolution do the authors use, finally?

- Page 12, line 10: Please add a bit more information about thin

`Opacity_Flag` product. How is this determined and how reliable is it? I understand that multiple profiles are combined. How is this done for the opacity flag?

**3.4.1 Data collocation**

- Page 12, line 25: "For this time period", referring to Sect. 3.3, but actually the time period is described in Sect. 3.4.

- Page 12, line 31: I'm confused. Higher on the same page the authors discuss how there are spatial resolutions at 5 km, 20 km, and 80 km. But now they seem to consider only 5 km. Then what is the relevance of the other spatial resolutions?

- Page 13, line 2: I believe the re-analysis also contains forecast variables at every hour, why not use those instead of interpolating the 6-hour time steps? Depending on what local time those correspond to a linear interpolation for surface temperature could introduce significant errors.

**3.4.2 Training data**

- Page 13, line 5: replace "millions" by "million"

- Page 13, line 11: do the authors use transparent as a synonym for "CALIOP signal did not get saturated"?

- Page 13, lines 15-19: the authors a huge training dataset, many orders of magnitude larger than in many other machine learning cases. Instead of duplicating certain cases, have the authors considered thinning the part of the state space where there are many cases, perhaps in a way similar to Chevallier (2016) https://nwpsaf.eu/downloads/profiles/profiles_91L.pdf ? That may provide a less biased dataset and (much) faster ANN training. Note that you are actually doing
this on page 15, line 10; first duplicating some points by a factor 4 and then using only 25% of the points is essentially thinning, depending on how the sub-selection is performed.

3.4.3 Validation data

• Page 13, line 28: what are the consequences of applying this balancing (or alternatively, as I propose above, thinning) to the training data but not to the internal validation data? This would mean that the statistical properties of the internal validation data differ from the ones for the training data. Can this introduce biased results?

3.5 Training

• Page 14, line 3: move "described in Sect. 2.3.2" to after "mini-batch gradient descent", because both backpropagation and mini-batch gradient descent are described there.

3.5.1 Training meta-parameters

• Page 14, Figure 2, legend: Replace "Tranparent" by "Transparent"

• Page 14, line 12: How is this random search performed? This is an optimisation problem and there are different ways of finding local or global minima.

• Page 14, lines 14-15: How is "best performing" defined? You have multiple metrics but it's not clear how those have been used exactly.

**3.5.2 MLP structure optimisation**

- Page 15, line 10: see my comment at page 13, lines 15–19

- Page 15, lines 18-19: Do you mean the differences between structures are very small, and/or the differences among the two trained for each structure?

- Page 31, Figure 3: It is hard to tell the differences between the performances. Could the authors add a figure showing the actual improvement (in %-point) between the network 3-64 and 1-16 and/or between the finally selected network and 1-16?

**4.1 Application**

- Page 17, line 16: replace "12.30" by "12:30"

- Page 32, figures 4(d)–(f): The colourmap chosen by the authors may not be optimal. As explained by Borland and Taylor (2007), the rainbow colour map and other colourmaps that are not perceptually uniform may be deceptive in their visual interpretation. The authors may wish to study the data using a perceptually uniform colourmap. Secondly, I do not understand why the colourmap in Figure 4(d) is different (opposite?) to the ones in 4(e) and 4(f). In this case, white has been used to indicate areas without cirrus clouds, so the colourmap should ideally not contain a colour similar to white (perhaps possibly at the low end of the IWP and IOT scales)

  Borland and Taylor (2007), Rainbow Color Map (Still) Considered Harmful, in: IEEE Computer Graphics and Applications ( Volume: 27, Issue: 2, March-April 2007 ), doi:10.1109/MCG.2007.323435

- Page 19, line 4: Replace "along side" by "alongside"

**4.2.2 Cirrus properties**

- Page 35, Figure 8: you are validating against CALIOP so the CALIOP measurement should be on the x-axis instead of on the y-axis. The same comment applies to Figures 10 and 11.

- Page 35, Figure 8: Why does the CALIOP scale go down to 4.0 km if the dataset excludes data with CTH < 4.5 km (poles) or 9.5 km (tropics)?

- Page 35, Figure 8, right panel: most of the lower part of the panel is empty. I think you can restrict the y-axis to -20% or so, and abandon the symmetry on both sides of the y=0-line. The same applies to Figures 10 and 11.

- Page 20, line 4: You might want to again point out here that the 4.5 km CTH in the dataset are all near the poles, so the problem for these pixels is actually more difficult than for others.

- Page 20, line 17 / Page 35, Figure 9: could you add a panel to Figure 9 showing the density of points that make up the statistics shown in Figure 9? You write in the text that those cases where there is a bias are relatively rare. Such a 2D histogram could show how rare.

- Page 36, Figure 10: comment at Figure 35, Figure 8 applies

- Page 36, Figure 10: caption should describe what the shaded area in the middle panel indicates. Currently this is only stated in the main text.

- Page 20, line 28: How is this (lack of correlation when $\mathrm{IOT_{CALIOP}} < 0.04$) apparent from Figure 10?

- Page 21, lines 19–21: $\mathrm{IOT_{CALIOP}}$ is your reference for the training and the validation. Why would a bias between $\mathrm{IOT_{CALIOP}}$ w.r.t. truth contribute to your error relative to $\mathrm{IOT_{CALIOP}}$?

- Page 36, Figure 11: comment at Figure 35, Figure 8 applies

- Page 22, line 4: I believe this result is not shown, so the authors may wish to indicate this for clarify (i.e. "(not shown)").

5 The cirrus life cycle with CiPS

- Page 22, line 12: I believe you mean Pyrenees, not Alps, or I'm confused.

- Page 22, line 15–20: Is this method an established technique or something that the authors developed? If the former, can you add a reference to a source containing more details? I realise it is not the main focus of the study but it would seem something the interested reader may wish to learn more details about.

- Page 22, line 33: replace "does also present" by "also presents".

- Page 23, line 5: Your cloud has its maximum area exactly at the time from which you tracked it forward or backward in time. Is this just a coincidence, or could it mean that your tracking is not entirely reliable? I'm a bit worried that the analysis in this paragraph may say more about your tracking method than about the cloud evolution, in particular for the surface area (not saying this is the case, but in theory it could be and therefore should be addressed or ruled out as an alternative explanation).

- Page 23, line 20: remove "though"

6 Conclusions

- Page 24, lines 9–10: there is a double negative here ("cannot ... neither ... nor"). I suggest to replace "neither ... nor" by "either ... or".

- Page 24, lines 11–14: the authors might want to briefly repeat the main points of why/how CiPS improves upon COCS.

Needs some lines on recommended future work / next steps.

Acknowledgements

Please expand/explain the acronyms in the acknowledgements, where they were not explained before (DLR, DAAD).

---

## Author Comment (AC1) · 26 Jun 2017

First of all we would like to thank the reviewer for taking the time to read and review our manuscript. The very thorough review with kind feedback and constructive comments certainly helps to improve and clarify the manuscript. Each comment from the reviewer is listed below along with the corresponding reply from the authors as well as possible changes in the manuscript (in italic font style).

**General comments**

The authors characterise the performance of CiPS in several ways. Did you look at its performance depending on underlying surface type? Snow surfaces are famously

difficult, but other surface aspects may be relevant as well.

*We have prepared a second manuscript where CiPS is further characterised. The underlaying surface type is one aspect, the presence of aerosols and liquid water clouds below the cirrus is another. As the reviewer implies, the retrievals are generally most uncertain over snow and ice. However this is only observed for IOT $< 0.5$ and IWP $< 10.0\,gm^{-2}$ respectively. For thicker cirrus the underlying surface type has a small effect. For further details, we encourage the reviewer to look out for the upcoming manuscript that will be submitted to AMT.*

The elephant in the room in many retrieval products, in particular those based on machine learning, is the uncertainty. Although the the paper provides characteristics on overall performance, is there any way to get an uncertainty estimate for a specific retrieval?

*This is another very good point. We have discussed several methods to derive uncertainties, however we have not found an approach that is more representative than the overall statistics in this manuscript. Below we list some uncertainty estimate approaches together with their limitations. 1) One approach is to train a second set of artificial neural networks (ANNs) to retrieve the uncertainty reported by CALIOP, this would however require that CiPS retrieves values that are identical or very close to those retrieved by CALIOP. As seen in the manuscript, this is not always the case. 2) A second approach is to run CiPS multiple times with small perturbations in the input data. This is however only representative if 100 % of the retrieval error can be attributed to noise and uncertainties in the input data. This is however not the case as most of the retrieval error of CiPS is likely to stem from the different sensitivities of SEVIRI and CALIOP. This is covered in the second manuscript, but then as a noise sensitivity analysis rather than an uncertainty estimate of CiPS. 3) A third approach is to train a second set of ANNs trained with the absolute difference between CiPS*

*and CALIOP (using the same collocation dataset used to train CiPS). The retrieved accuracy would however be a statistical uncertainty learned using several training points and we consider the added value, in comparison to the characteristics on the overall performance presented here, to be small. So rather than estimating individual uncertainty measures of each retrieval we focus on performing a detailed characterisation of CiPS that we present here and in the second manuscript.*

Is there, or is there planned to be, a publicly available data product based on CiPS, so that people can download the data and explore it on their own? I think there should be people interested in using it.

*At the moment there is no publicly available CiPS product. Up to know CiPS has been applied to case studies and we do not have a long time series of CiPS data. We thank the reviewer for the comment and will look at the possibilities in making the data publicly available. In the meantime, users are encouraged to contact the authors if they are interested in using the data, making inter-comparisons etc.*

Could the approach be extended to other imagers than SEVIRI, as long as those have an overlap with CALIOP to be trained with? Or are the properties of SEVIRI (footprint size? scan speed? channels?) essential for CiPS to work?

*In general it should be possible to extended this method to an imager that regularly overlaps with CALIOP and has multiple channels within the thermal infrared spectral range $\approx 6.2 - 13.4 \mu m$. As the reviewer points out, one limiting factor could be the spatial resolution. SEVIRI has a spatial resolution of $3 \times 3\,km^2$ at nadir which increases to approx. $4 \times 5\,km^2$ in mid-latitudes. This agrees well with the $5\,km$ spatial resolution of CALIOP that we use. Using instruments with a higher spatial resolution like MODIS ($1 \times 1\,km^2$ at nadir) or ABI/AHI ($2 \times 2\,km^2$ at nadir) it is possible that the imager data would have to be re-gridded to a coarser resolution in order to better agree with*

*the spatial resolution of the training reference CALIOP data, especially for the high resolution MODIS data. The scan speed should not be a limiting factor for developing a CiPS-like algorithm for another geostationary imager. If the scan speed is faster than the 15 min of SEVIRI, it would only be an advantage and if the scan speed would be slower than 15 min one could implement a threshold saying that the acquisition time difference between CALIOP and the imager can not be larger than 7.5 min, which is the maximum time difference between CALIOP and SEVIRI.*

**Specific comments**

**Abstract**

Page 1, lines 1–2: replace "one of the largest uncertainties" by "one of the largest sources of uncertainty", and replace "they" by "their physical properties"

*Revised*

Page 1, line 8: after 71, add %

*Revised*

**1. Introduction**

Page 3, line 29: add "piece of" before "information"

*Revised*

Page 3, line 32: please explain acronym LES (I guess this is large eddy simulation in the context of a cloud resolving model), which is missing in the text and in Appendix A.

As this acronym appears to be used only once in the paper, I suggest just writing it out and avoiding the acronym altogether.

*Correct, LEM stands for large eddy simulation in this context. This has been revised according to the suggestion by the reviewer.*

**2.1 SEVIRI**

Please expand this paragraph with:

- longitude above which SEVIRI is located (finally described on page 12, line 22)

- total range of field of view of the disk SEVIRI can observe

*Revised. The following sentence has been added to the paragraph "SEVIRI is positioned at $0°$ E (operational service) and has an excellent view of the Earth from its remote location, with a spatial coverage from approx. $80°$ W to $80°$ E and $80°$ S to $80°$ N."*

**2.2 CALIOP**

Page 4, line 18: This usage of the word "frequency" is potentially confusing, maybe write that it measures 20.16 times per second or every 49.6 ms (when I see the word frequency I think of electromagnetic frequency).

*Revised. The sentence now reads as follows: "By emitting approx. 20 laser pulses per second, a 70 m footprint is produced every 335 m on the Earth's surface, resulting in curtains of attenuated backscatter profiles along the CALIPSO track."*

Page 4, line 24: please explain acronym IOT

*Revised. The acronym IOT is defined already in the introduction, but since it's not as*

*household as for example IWP, an additional explanation is a good idea.*

Page 4, line 26: This line has some typesetting issues: CPL should be explained at first use, and the formatting of the citation is incorrect.

*Revised.*

**2.3.2 Learning through backpropagation**

Page 6, line 16: replace "a" by "an".

*Revised.*

**2.4 Validation metrics**

Page 8, equation 5: this MPE metric is risky because over- and underestimates can cancel out each other. I realise that is why the authors also look at MAPE but I think this risk should be explicitly pointed out.

*Revised. The following sentence has been added: "When calculating the MPE, over- and underestimations can cancel out each other, potentially leading to zero MPE (bias) even if the magnitude of the errors is large. Therefore the MAPE has been considered as well."*

**3 CiPS**

Page 9, line 4: remove "though"

*Revised.*

Page 9, line 5: remove "the"

*Revised.*

**3.1 Multiple artificial neural networks**

Page 9, line 10: Remove "decimal" before "number". I don't think the authors actually mean a decimal number as defined by IEEE 754-2008, presumably it's a regular binary in their software implementation.

*We do mean a 32-bit floating point number in the interval (0,1). Since the ANN uses a continuous activation function, the classification ANN does not retrieve a binary number directly, but a decimal number between 0.0 and 1.0 that can be seen as a cirrus probability. Therefore a threshold has to be defined in order to obtain a binary cirrus cloud flag (see Sect. 3.6 in the discussion manuscript). The sentence has been clarified in the manuscript and now reads as follows: "Due to the continuous activation function used by the ANN (Sect. 2.3.1), the retrieved value of the CCF neuron is a real number in the interval (0,1) represented by a 32-bit floating point number."*

Page 9, line 23-24: Does CALIOP (reliably) identify when it is saturated?

*CALIOP is considered saturated when no backscatter signal can be distinguished from the background signal level. It could be possible that there is still a small backscatter signal left at this point, even if CALIOP can not distinguish it from the background signal. This should however not have any impact on the retrieved properties, since the altitude at which the last distinguishable signal was observed is recorded. Please see the reply to the comment regarding Page 12, line 10 (in the discussion paper) for additional information.*

Page 10, line 2: The authors refer to "photon counts" but I don't expect SEVIRI actually counts photons. The digital count level is probably rather a conversion from a voltage. I assume the authors use brightness temperatures already calibrated elsewhere, so I suggest to cite the relevant paper or technical report if available.

*We thank the reviewer for pointing this out, this was indeed not a correct description. The manuscript now reads as follows: "Brightness temperatures from all thermal channels of SEVIRI except for the ozone channel at $9.7\,\mu m$ are used. The brightness temperatures are calculated according to EUMETSAT (2012)."*

*EUMETSAT: The Conversion from Effective Radiances to Equivalent Brightness Temperatures,*
*EUM/MET/TEN/11/0569, 2012.*

**3.2.1 Brightness temperatures from SEVIRI**

Page 10, line 8: It would be useful to remind the reader to what surface area 19x19 pixels$^2$ corresponds (57x57 km$^2$?)

*Revised.*

**3.3 Output data: cirrus properties from CALIOP**

Page 12, line 3: Which spatial resolution do the authors use, finally?

*This is indeed a bit tricky to understand. We use the product with a reported spatial resolution of 5 km. But to detect faint cirrus and aerosol layers, the CALIOP team has to average over several consecutive 5 km profiles in order to get a sufficiently high signal-to-noise ratio. This means that in the 5 km cloud layer product, some cirrus were detected using a spatial resolution of 20 or even 80 km. In such a case the 5 km layer*

*product will have 4 or 16 consecutive bins where the cirrus properties are identical. The additional spatial resolutions of 20 and 80 km can be seen as "background resolutions" used by the CALIOP team. This has been clarified by extending the paragraph, which now reads as follows: "Even though the cloud and aerosol layer product are reported with a spatial resolution of 5 km, two additional coarser resolutions of 20 and 80 km are used to detect the cloud and aerosol layers reported in the 5 km products (Vaughan et al., 2009). At a spatial resolution of 5 km, the signal-to-noise ratio of a faint cirrus or aerosol layer is usually too weak to be distinguished from the clear-sky atmospheric signal. By averaging 4 or 16 consecutive 5 km profiles the signal-to-noise ratio is increased, which allows for detection of very thin cirrus and aerosol layers. For example if a thin cirrus cloud with an optical thickness of 0.1 and a top altitude of 10 km is identified only when 16 consecutive 5 km profiles are averaged (80 km spatial resolution), 16 consecutive bins in the L2 5 km cloud layer data will report an optical thickness of 0.1 and a top altitude of 10 km."*

Page 12, line 10: Please add a bit more information about thin Opacity_Flag product. How is this determined and how reliable is it? I understand that multiple profiles are combined. How is this done for the opacity flag?

*Just to be clear, the opacity flag does not tell if an aerosol or cloud layer is opaque in the normal sense of the term. Instead it gives the information whether the CALIOP backscatter signal was completely attenuated within a detected layer (i.e. became indistinguishable from the background signal level). Therefore a rather thin cirrus or aerosol layer can be classified as opaque if most of the signal was backscattered at higher levels.*

*An aerosol or cloud layer is considered opaque if it is the lowermost layer detected by CALIOP and no surface return is observed below that layer. The surface return is identified by looking at a digital elevation model (DEM), the width of the feature (the*

*surface return is comparably narrow) and the magnitude of the backscatter. Hence the opacity classification for cirrus clouds should be one of the more accurate as the base altitude of cirrus cloud layers is unlikely to be at the surface level and hence be confused with a surface return.*

*As the reviewer implies, the CALIOP retrieval scheme uses profiles with a spatial resolution of 5 km (the fundamental resolution of the retrieval scheme), 20 km (average of 4 consecutive 5 km profiles) and 80 km (average of 16 consecutive 5 km profiles). The base altitudes of all features detected at the three spatial resolutions are then compared to the corresponding maximum penetration depth (MPD). The MPD reports the base altitude of the lowest feature with a spatial resolution of 5 km. If the lowest feature is not detected at 5 km, the base altitude retrieved at the coarser resolution is used. For the 5 km profiles the opacity identification is straight forward; if the MPD is lower than the feature's base altitude, the feature is classified as transparent. Features detected at a coarser resolution are classified as transparent if at least 50 % of the corresponding MPDs within the 20 or 80 km distance are lower than the corresponding feature base altitude.*

*The following sentences have been added to Sect. 3.3: "The Opacity_Flag gives the information whether the CALIOP backscatter signal was completely attenuated within a detected layer. During the CALIOP retrieval, a cirrus cloud layer is classified as opaque if it is the lowermost layer and not identified as a surface return (Vaughan et al., 2005). A digital elevation model is partly used to identify surface returns, meaning that high cirrus clouds should not be falsely classified with respect to transparency. Cirrus cloud layers detected at the coarser 20 km or 80 km resolutions are classified as transparent if the corresponding base altitude is higher than the lowermost detected feature in at least 50 % of the 4 or 16 consecutive 5 km profiles that constitute the 20 km and 80 km averages."*

*Vaughan, M. A., Winker, D. M., and Powell, K. A.: CALIOP Algorithm Theoretical Basis*

*Document Part 2: Feature Detection and Layer Properties Algorithms, PC-SCI-202 Part 2, 2005.*

**3.4.1 Data collocation**

Page 12, line 25: "For this time period", referring to Sect. 3.3, but actually the time period is described in Sect. 3.4.

*The reference was meant to refer to the CALIOP data and the corresponding quality screening described in Sect. 3.3. To avoid mis-interpretations the cross-reference has been removed.*

Page 12, line 31: I'm confused. Higher on the same page the authors discuss how there are spatial resolutions at 5 km, 20 km, and 80 km. But now they seem to consider only 5 km. Then what is the relevance of the other spatial resolutions?

*See response above. We hope that the additional sentences help understanding the meaning of the different resolutions related to the CALIOP 5 km layer products.*

Page 13, line 2: I believe the re-analysis also contains forecast variables at every hour, why not use those instead of interpolating the 6-hour time steps? Depending on what local time those correspond to a linear interpolation for surface temperature could introduce significant errors.

*As far as we know the forecast data contained in the ECMWF ERA-Interim dataset is available every three hours. The reviewer still has a very good point. We use the re-analysis data in order to avoid errors propagating in time due to errors in the forecast model. As the reviewer points out, the linear interpolation introduces errors as well and for future work one could make a sensitivity analysis between the re-analysis*

*and forecast fields in order to see which method is superior: using the more accurate re-analysis data with a coarser temporal resolution or the less accurate forecast data with a higher temporal resolution.*

**3.4.2 Training data**

Page 13, line 5: replace "millions" by "million"

*Revised.*

Page 13, line 11: do the authors use transparent as a synonym for "CALIOP signal did not get saturated"?

*Yes. The sentence has been clarified and now reads as follows: "Furthermore, the IOT/IWP ANN is trained only with collocations containing transparent cirrus clouds, where the CALIOP signal was not saturated such that the true, rather than the apparent, IOT and IWP could be retrieved.".*

Page 13, lines 15-19: the authors a huge training dataset, many orders of magnitude larger than in many other machine learning cases. Instead of duplicating certain cases, have the authors considered thinning the part of the state space where there are many cases, perhaps in a way similar to Chevallier (2016) https://nwpsaf.eu/downloads/profiles/profiles_91L.pdf? That may provide a less biased dataset and (much) faster ANN training. Note that you are actually doing this on page 15, line 10; first duplicating some points by a factor 4 and then using only 25 % of the points is essentially thinning, depending on how the sub-selection is performed.

*Yes, we had a thinning approach in mind, but in the end we decided not to do it. Since we knew that the most difficult retrievals for SEVIRI would be the thinnest cirrus*

*clouds, we did not want to remove potentially valuable training information by thinning the dataset. Therefore we chose to increase the weight of the comparably rare cases instead by adding duplicates. It is possible (perhaps likely) that the approach proposed by the reviewer would have reduced the training time for CiPS, without reducing the accuracy. We will keep this approach in mind for further developments. For an unbalanced training dataset, where common points are expected to be as easy/difficult to retrieve as the other points, we fully agree that such an approach would be very efficient, especially with regards to training time.*

*The 25 % subset used for the first stage of the training was selected randomly, such that the weight of any cirrus type (thin, thick, high, low, etc.) during the training remained the same.*

**3.4.3 Validation data**

Page 13, line 28: what are the consequences of applying this balancing (or alternatively, as I propose above, thinning) to the training data but not to the internal validation data? This would mean that the statistical properties of the internal validation data differ from the ones for the training data. Can this introduce biased results?

*This is also a good point, but our idea of the balancing is to give the rare points a stronger weight during the training, otherwise their contribution to the weight updates might be too weak to learn the relationship. The internal validation dataset is used to monitor the error against independent data in order to determine when the training shall be stopped. If we would balance both the training and the internal validation dataset we would train until our statistics is as similar as possible to the statistics of the balanced internal validation dataset and the ANNs would thus learn according to the wrong statistics. Therefore we leave the internal validation dataset unbalanced in order to learn according to the "true" cirrus cloud statistics observed by CALIOP.*

**3.5 Training**

Page 14, line 3: move "described in Sect. 2.3.2" to after "mini-batch gradient descent", because both backpropagation and mini-batch gradient descent are described there.

*Revised.*

**3.5.1 Training meta-parameters**

Page 14, Figure 2, legend: Replace "Tranparent" by "Transparent"

*Revised.*

Page 14, line 12: How is this random search performed? This is an optimisation problem and there are different ways of finding local or global minima.

*This has been clarified according to the reviewer's comment. The manuscript now reads as follows "To find the optimal values for each meta-parameter, a random search according to Bergstra and Bengio (2012) is performed within intervals chosen based on expert knowledge. Sets of meta-parameters are randomly drawn from the pre-defined intervals and used to train corresponding sets of ANNs. Assuming an infinite number of samples, this procedure can be regarded as a global optimization technique. The optimal set of meta-parameters is defined as the one that minimises the mean squared error (MSE) between the ANN and reference data using an independent test set.".*

*Bergstra, J. and Bengio, Y.: Random search for hyper-parameter optimization, Journal of Machine Learning Research, 13, 281–305, 2012.*

Page 14, lines 14-15: How is "best performing" defined? You have multiple metrics but

it's not clear how those have been used exactly.

*The mean squared error between the ANN and independent reference data is used for this purpose. This has been clarified and the sentence now reads as follows: "For both the classification and regression tasks a learning rate around 0.05 and momentum around 0.99 is found to provide ANNs with the lowest MSE against the independent reference data."*

**3.5.2 MLP structure optimisation**

Page 15, line 10: see my comment at page 13, lines 15-19

*See response to comment regarding page 13, lines 15-19.*

Page 15, lines 18-19: Do you mean the differences between structures are very small, and/or the differences among the two trained for each structure?

*It refers to the differences between the two networks trained for each structure. This has been clarified in the manuscript: "The differences between the two networks trained for each structure are however very small."*

Page 31, Figure 3: It is hard to tell the differences between the performances. Could the authors add a figure showing the actual improvement (in %-point) between the network 3-64 and 1-16 and/or between the finally selected network and 1-16?

*We thank the reviewer for the suggestion, this does indeed show the actual improvements in a much better way. Three sub-figures have been added to Fig. 3 showing the difference between the seven different ANN structures and the least complex one (1-16) for the cirrus cloud detection and CTH/IOT retrieval respectively.*

*Furthermore the following sentences have been added to the text in conjunction to Fig. 3: "Figure 3d shows the difference in POD between each structure and the least complex structure having one hidden layer and 16 hidden neurons (1-16). Similarly, Fig. 3e and Fig. 3f show the difference in MAPE between each structure and the least complex one for the CTH and IOT retrievals respectively.". Also the figure caption has been extended in a similar way. As the additional sub-figures give a more detailed overview of the differences between the structures, the paragraph starting at page 15 line 30 and ending at page 16 line 3 has been revised and now reads as follows: "In all cases, already small networks produce reasonable results. In many cases differences between structures are not very large. Nevertheless, we also see that larger ANNs can always solve the problems in a more accurate way and especially for the cirrus cloud detection it is beneficial to either use more hidden neurons or add more hidden layers rather than using a simple structure with one hidden layer and 16 hidden neurons (1-16). Using two or three hidden layers with 64 hidden neurons each (2-64, 3-64) yields a POD that is up to 8 percentage points higher compared to one hidden layer with 16 hidden neurons (1-16). Similarly, a structure with three hidden layers and 16 hidden neurons (3-16) yields a POD that is up to 5.5 percentage points higher compared to the structure with one hidden layer and 16 hidden neurons (1-16). Although three hidden layers with 64 neurons each (3-64) offers the highest accuracy for all cases, such a complex structure processes the data significantly slower by a factor 8 or 6 compared to the smaller structures with 2 or 3 hidden layers and 16 neurons per layer. For the IOT retrieval, a larger ANN is mostly beneficial for the thinner cirrus and the MAPE with respect to CALIOP seems to be saturated and hardly improvable for $IOT_{CALIOP} > 0.1$ using this approach and training data. For the sub-visual cirrus, the MAPE with respect to the CALIOP reference IOT is up to 13 percentage points lower using two hidden layers instead of one hidden layer with 16 hidden neurons each. For the CTH retrieval, only marginal improvements in the MAPE with respect to CALIOP ($\approx 0.1 - 0.5$ percentage points) are observed using the more complex structures in comparison to the least complex one (1-16). Only for the*

*lowermost clouds ($CTH_{CALIOP} < 6.0$ km) the advantage of using more hidden layers and neurons is more evident."*

**4.1 Application**

Page 17, line 16: replace "12.30" by "12:30"

*Revised.*

Page 32, figures 4(d)-(f): The colourmap chosen by the authors may not be optimal. As explained by Borland and Taylor (2007), the rainbow colour map and other colourmaps that are not perceptually uniform may be deceptive in their visual interpretation. The authors may wish to study the data using a perceptually uniform colourmap. Secondly, I do not understand why the colourmap in Figure 4(d) is different (opposite?) to the ones in 4(e) and 4(f). In this case, white has been used to indicate areas without cirrus clouds, so the colourmap should ideally not contain a colour similar to white (perhaps possibly at the low end of the IWP and IOT scales) Borland and Taylor (2007), Rainbow Color Map (Still) Considered Harmful, in: IEEE Computer Graphics and Applications ( Volume: 27, Issue: 2, March-April 2007 ), doi:10.1109/MCG.2007.323435

*We thank the reviewer for this very interesting comment. We were not aware of the problem with rainbow colourmaps. We have replaced the previous colourmap with the perceptually uniform Viridis colourmap.*

*We used another colourmap (ranging from red to blue) for the cloud top height as it might be more intuitive for the reader if lower/warmer clouds are represented as red and higher/colder cirrus as blue. With the new Viridis colourmap this is not a problem and the same colourmap is now used for Fig. 1d-f.*

Page 19, line 4: Replace "along side" by "alongside"

*Revised.*

**4.2.2 Cirrus properties**

Page 35, Figure 8: you are validating against CALIOP so the CALIOP measurement should be on the x-axis instead of on the y-axis. The same comment applies to Figures 10 and 11.

*Revised.*

Page 35, Figure 8: Why does the CALIOP scale go down to 4.0 km if the dataset excludes data with CTH < 4.5 km (poles) or 9.5 km (tropics)?

*This only applies to the COCS algorithm, not the CiPS algorithm that we introduce here. CiPS does not have a lower or upper limit. In Sect. 3.3 we write "The improved quality of the V3 CALIOP products allows us to omit the filtering processes used for COCS (see Sect. 2.5)".*

Page 35, Figure 8, right panel: most of the lower part of the panel is empty. I think you can restrict the y-axis to -20% or so, and abandon the symmetry on both sides of the y=0-line. The same applies to Figures 10 and 11.

*The reviewer has a very good point, this has been revised. The y-axis now covers the interval [-50,100]. The reviewer is right that we could restrict the y-axis to -20 %, but to better match the second manuscript, we choose to restrict the y-axis to -50 %.*

Page 20, line 4: You might want to again point out here that the 4.5 km CTH in the

dataset are all near the poles, so the problem for these pixels is actually more difficult than for others.

*The reviewer is absolutely correct. The following sentence has been added to the manuscript "Furthermore, this type of low cirrus/icy clouds are found in the polar regions (see Fig. 9b), where the retrieval conditions for SEVIRI are more challenging with larger viewing zenith angles and pixel sizes."*

Page 20, line 17 / Page 35, Figure 9: could you add a panel to Figure 9 showing the density of points that make up the statistics shown in Figure 9? You write in the text that those cases where there is a bias are relatively rare. Such a 2D histogram could show how rare.

*We thank the reviewer for the good suggestion. Such a figure has been added. Furthermore the following sentences have been added to the text in conjunction to Fig. 9: "Figure 9b shows the corresponding occurrences of the points that make up the statistics shown in Fig. 9a. Please remember that the validation dataset is a random subset of CALIOP data collected over a time period of almost six years and hence represents the natural latitudinal distribution of cloud top heights.". Also the figure caption has been extended in a similar way. The sentence starting at page 20 line 15 and ending at page 15 line 17 has been revised and now reads as follows: "From Fig. 9b it is clear that the situations with higher errors and stronger biases (MPE $> 20$ %) are comparably rare and that $CTH_{CiPS}$ is unbiased for the more frequent combinations of $CTH_{CiPS}$ and latitude.".*

Page 36, Figure 10: comment at Page 35, Figure 8 applies

*Revised. The CALIOP data is now presented on the horizontal axes in Fig. 10a,b and the y-axis in Fig. 10c has been restricted to -100 %.*

Page 36, Figure 10: caption should describe what the shaded area in the middle panel indicates. Currently this is only stated in the main text.

*Revised.*

Page 20, line 28: How is this (lack of correlation when $IOT_{CALIOP} < 0.04$) apparent from Figure 10?

*For $IOT_{CALIOP} < 0.04$, CiPS clearly overestimates the IOT with a wide spread for the estimates leading to a poor accuracy/correlation. We consider that this is clear from the left panel in Fig. 10.*

Page 21, lines 19-21: $IOT_{CALIOP}$ is your reference for the training and the validation. Why would a bias between $IOT_{CALIOP}$ w.r.t. truth contribute to your error relative to $IOT_{CALIOP}$?

*The reviewer is correct, the information in that sentence is not correct. As we train and validate CiPS with the same (but independent) biased data we should not see any bias with respect to the truth during the validation against CALIOP. We thank the reviewer for pointing this out. The corresponding sentence has been remove from the manuscript.*

Page 36, Figure 11: comment at Page 35, Figure 8 applies

*Revised. The CALIOP data is now presented on the horizontal axes in Fig. 11a and the y-axis in Fig. 11b has been restricted to -100 %.*

Page 22, line 4: I believe this result is not shown, so the authors may wish to indicate this for clarify (i.e. "(not shown)").

*Revised.*

**5 The cirrus life cycle with CiPS**

Page 22, line 12: I believe you mean Pyrenees, not Alps, or I'm confused.

*Thanks for the comment, this was indeed not very clear. We did however mean the Alps, in the sense that the cirrus cloud later detected south of the Pyrenees originated from the outflow of an orographic cirrus located south of the Alps. To clarify this, the sentence "On September 26, 2014 an orographic cirrus cloud was observed south of the Alps (see Fig. 12a)" has been removed.*

Page 22, line 15-20: Is this method an established technique or something that the authors developed? If the former, can you add a reference to a source containing more details? I realise it is not the main focus of the study but it would seem something the interested reader may wish to learn more details about.

*This is something that we have developed, but there are similar techniques in the literature. A reference has been added.*

Page 22, line 33: replace "does also present" by "also presents".

*Revised.*

Page 23, line 5: Your cloud has its maximum area exactly at the time from which you tracked it forward or backward in time. Is this just a coincidence, or could it

mean that your tracking is not entirely reliable? I'm a bit worried that the analysis in this paragraph may say more about your tracking method than about the cloud evolution, in particular for the surface area (not saying this is the case, but in theory it could be and therefore should be addressed or ruled out as an alternative explanation).

*We wanted to initialise the tracking before the cirrus starts to split into several patches. But that we selected the point of maximum area is a coincidence. Based on the reviewer's comment, we validated the tracking method by initiating the tracking at two different times, 08:00 UTC and 12:00 UTC. Only marginal differences were observed, arising from the fact that some small cirrus patches, that in the end form the tracked cirrus, might be temporarily missed (which is exactly the reason why we selected a starting time when there were few small single patches close to the cirrus). The following lines have been added to the manuscript: "Triggering the tracking 2 h before/after the starting time presented here (10:00 UTC) results in only marginal differences ($< 5.0$ % in horizontal area, not shown here), as some small cirrus patches that in the end form the tracked cirrus might be temporarily missed. This validates the robustness of the tracking method".*

Page 23, line 20: remove "though"

*Revised.*

**6 Conclusions**

Page 24, lines 9-10: there is a double negative here ("cannot ... neither ... nor"). I suggest to replace "neither ... nor" by "either ... or".

*This has been clarified and the sentence now reads as follows: "This information is very important to discern thin cirrus, for which CiPS works very well, from thicker*

*clouds where neither CiPS nor CALIOP can capture the complete IOT and IWP."*

Page 24, lines 11-14: the authors might want to briefly repeat the main points of why/how CiPS improves upon COCS.

*Revised. The following sentences have been added: "Improvements with respects to COCS can be attributed to several factors. 1) We use new input data including the modelled surface skin temperature and the regional maximum and average brightness temperatures. 2) The training meta-parameters and ANN structures have been thoroughly investigated and optimised for CiPS. 3) The training of CiPS was more rigorous, with mini-batch learning rather than stochasitc learning as well as a tuning phase with gradually increasing batch size and gradually decreasing learning rate and momentum. Furthermore an internal validation dataset was used during the training of CiPS in order to monitor the accuracy and avoid overfitting. 4) The use of the more accurate V3 CALIOP data allowed us to omit the CTH filtering used for COCS, leading to a more accurate CTH retrieval by CiPS. 5) CiPS utilises multiple ANNs. COCS uses one single ANN trained with cirrus covered as well as cirrus free pixels. On the contrary, the CiPS ANNs that retrieve the CTH, IOT, IWP and OPF were trained exclusively with cirrus covered pixels, resulting in lower retrieval errors of CiPS. The larger retrieval errors of COCS for thin cirrus clouds also affects the IOT dependent cirrus cloud detection of COCS, with both a lower POD and a higher FAR compared to CiPS.*

Needs some lines on recommended future work / next steps.

*The following lines have been added to the conclusions: "As a next step, the CiPS retrievals will be further characterised with respect to the underlying surface type and the presence of aerosol layers and liquid water clouds below the cirrus. Constant developments and improvements of the CALIOP cirrus cloud retrievals also opens the*

*door for further improvements of CiPS. Another aspect of improvement would be to introduce new input data, for example temperature and humidity profiles and surface emissivity. One could also investigate the usefulness of a more rigorous balancing of the training dataset in order to reduce the number of training points without loosing any unique information."*

**Acknowledgements**

Please expand/explain the acronyms in the acknowledgements, where they were not explained before (DLR, DAAD).

*Revised.*

---

## Referee Comment (RC2) · Anonymous Referee #3 · 12 Jul 2017

The manuscript presents new inversions of MSG/SEVIRI data, providing information on cirrus clouds. The retrieval products are cloud top height, optical thickness and ice water path. As the authors describe, there exists a lack of measurements of cirrus properties. The best cirrus data are today provided by CALIOP and CloudSat, that are both active instruments flying together in a sun synchronous orbit and have both swath widths of about 2 km. This gives poor spatial and temporal coverage, and complementing retrievals by passive instruments are required. Making use of a geostationary instrument, such as MSG/SEVIRI, limits the geographical coverage but excellent diurnal coverage can obtained. The authors also selected to just use infrared observations to obtain 24 h coverage. In addition, SEVIRI provides 15 min resolution. Accordingly, the manuscript has a good justification and the topic fits well with AMT.

[Figure]

The authors selected to apply artificial neural networks (ANNs), and a large fraction of the manuscript describes the procedure for selecting net topology and training approach. I would say that ANNs today are used quite broadly and this part is too detailed (more below). On the other hand, the core element in the training dataset is CALIOP retrievals and there is basically no discussion of the limitations and accuracy of those retrievals. This information is required as all limitations in the training dataset are inherited by the ANN retrievals. In addition, there is no motivation to why CALIOP-only retrievals were selected for the training.

Further, as the other referee, I note the lack of a case specific error characterisation. In fact, there is not even a proper general error characterisation as the errors inherited from CALIOP are not considered. There should also be errors caused by the collocation procedure, as discussed below. In my opinion, this is not satisfactory. However, this is a general issue for ANN retrievals, and it is probably easy to find similar examples published recently. I happen to notice that the authors have submitted a new manuscript, with a title indicating that an extended error analysis now is at hand. I leave it to the editor to judge the overall situation, and potentially consider if these two manuscripts should be joined into a single manuscript.

General comments:

As indicated above, Section 3 could be shortened considerable. In fact, I think the section would become much more clear if the final net topology and training are simply presented "as given". If there is any general experience to draw from the tests performed to reach the final configuration, summarise these separately. The present detailed description of the tests just obscures the final outcome, and it is very hard to extract if there is any experience of general interest.

Some comments on terminology used around the ANN training. In "machine learning" one is usually supposed to work with three datasets: training, validation and test set. The training set should be used to train one or several methods, validation set should

be used to select the best one, and finally the test set should be used to evaluate the final system. In the manuscript the authors seem to mix up these sets. They use the validation set to monitor the training of the ANNs and then use the test set to select the best one and evaluate the performance. This is probably not critical for large datasets, nevertheless from a conceptual point of view this is not very nice. The authors refer to those datasets as training data, internal validation data, and (final) validation data, respectively.

Since this is also relevant for the follow-up paper, I suggest the authors create an additional test set that is used exclusively for evaluation.

The authors should reflect upon if using CALIOP alone is the best choice for training data. Why not use some combined CloudSat and CALIOP retrievals, such as DAR-DAR? As far as I understand, the CALIOP lidar signal is quickly attenuated and I assume that the SEVIRI IR channels have a deeper penetration into the clouds. That is, CALIOP alone does not span a sufficient range of IWP. This problem should vanish if using e.g. DARDAR for training. This comment is a hint, no demand for redoing the work.

However, this touches upon the comment made above, that the errors of the CALIOP retrievals must be reported. These errors propagate directly into errors in the ANN product. As the test dataset is taken from the same CALIOP retrievals, the inherent CALIOP errors are not revealed. Conservative, quantitative, values on the dynamic range (i.e. coverage of optical thickness and IWP) and accuracy of the CALIOP product used for training shall be given.

Further, there are also errors originating in the collocation procedure. Probably most important is the fact that CALIOP has a swath smaller than the resolution of SEVIRI. This results in that CALIOP covers only a part of the SEVIRI footprint, and this gives an additional uncertainty in the empirical relationships between CALIOP retrievals and SEVIRI measurements that the ANN is trained to represent. In any case, there are no

comments at all of possible errors caused by imperfections in the collocation procedure.

The points raised in the last two paragraphs, are they considered in the new manuscript targeting errors?

Specific Comments:

p 5, l 23: Also the bias neurons need to be assigned the correct values.

Sec 3.3: The section fails to clearly report what spatial resolution that is applied. For me this became clear first when reaching p 12, l 31. As 5 km anyhow is used, is the main discussion in Sec 3.3 actually needed? It seems to refer to an older version. That is, this section could be shortened.

p 13, l 24: This is the only place where the authors mention the activation functions of the networks. This information should not be given below "Training data", but as indicated below. Further, why don't the authors use identity activation functions on the output layers of the regression ANNs instead of re-scaling, which would be the more common approach? This could also have an effect on the learning of extreme values since the gradient vanishes at both limits of the output range.

p 15, l 1-10: When introducing the structure of the networks the authors should mention which activation functions are used in the hidden layers and the output layer.

p 15, l 28-29: 1 CPU@3.4 GHz does not describe the computer sufficiently especially since ANN inference is highly parallelizable. The authors should at least give number of cores and processor model.

p 16, l 25: Figure 3 does not seem compare the performance after the final training to the performance before, so the reference here seems pointless.

p 17, l 5-13: The detection threshold for the CCF is a parameter of the classification ANN and is thus prone to overfitting. Its value should not be determined based on the performance on the test set. This touches upon the general comment above. Also,
a plot of the POD against FAR for different thresholds would be good as it gives an additional perspective on the performance of the classifier.

p 18, l 4: 'the values corresponds' should be 'the values correspond'

p 18, l 9: Also here the authors claim the test data was excluded from the training but it seems that it has been used for the selection of the network structure and meta parameters.

p 19, l 3 - 6: See comment p 17, l 5 - 13

p 21, l 5: 'might seems high' should be 'seem'

p 21, l 7 - 14: The authors should explain in more detail what they mean with uncertainty and solution and/or provide a reference for their claims on the behaviour of ANNs.

p21, l 15 - 21: Isn't that the reason for the 'uncertainty' mentioned in the paragraph (see comment p21, l 7 - 14) in the training data? (I.e. low signal to noise ratio for thin clouds?)
* * *

---

## Author Comment (AC2) · 8 Aug 2017

**Reply to Anonymous Reviewer #3**

We are very grateful to the reviewer for reading and reviewing our manuscript. Considering the reviewer's constructive comments clearly improved the quality of the manuscript. Each comment from the reviewer (roman style) is listed below along with the corresponding reply from the authors (in italic font style) as well as possible changes in the manuscript (in blue italic font style).

First we briefly want to address the general question raised by the reviewer on whether this manuscript could be joined with a recently submitted manuscript where the CiPS

algorithm is characterised. Even though the two manuscripts are thematically related we think it is reasonable to split them. The first manuscript (this one) describes the development of a new tool, CiPS, for the passive remote sensing of cirrus clouds from MSG/SEVIRI together with a comprehensive evaluation of its overall performance, a comparison to a similar retrieval, COCS, and an application to the cirrus life cycle. The overall CiPS and COCS performance is assessed against CALIOP, that represents our "truth", and the errors of all CiPS/COCS retrieved variables have been investigated as a function of the corresponding CALIOP quantities. Furthermore, the geographical distribution of the False Alarm Rate (FAR) and the latitudinal distribution of Cloud Top Height (CTH) errors are also discussed. The comparison to COCS enables to quantify the improvements achieved with CiPS. Thus, this paper contains in our opinion all essential information about CiPS and resembles, in its structure, the paper by e.g. Kox et al. (2014) and Holl et al. (2014). Altogether the first paper is more than 20 pages long.

In the second manuscript additional aspects are investigated, including the effect of the underlying surface type and the presence of liquid water clouds and aerosols below the cirrus on the cirrus cloud retrieval, the sensitivity to radiometric noise from SEVIRI, the relative weight of the single input variables, and the characterisation of the errors of the CiPS output quantities IOT (Ice Optical Thickness) and CTH as a function of both CALIOP IOT and CTH simultaneously. These aspects are investigated for CiPS alone since they represent a detailed characterisation of CiPS and are less important and less interesting for the "older and less accurate" COCS. The CiPS user can learn from this second paper additional useful information abut the algorithm and its performance in all these respects. Nevertheless, even though those aspects are investigated for CiPS alone, we consider this knowledge interesting for the broader cirrus cloud remote sensing community and it does not fit the topic of the first paper which, in a broad sense, is the development of CiPS. Altogether the second paper is around 20 pages long as well.

Kox, S., Bugliaro, L., and Ostler, A.: Retrieval of cirrus cloud optical thickness and top altitude from geostationary remote sensing, Atmos. Meas. Tech., 7, 3233-3246, 2014.

Holl, G., Eliasson, S., Mendrok, J., and Buehler, S. A.: SPARE-ICE: Synergistic ice water path from passive operational sensors, J. Geophys. Res. Atmos., 119, 1504–1523, 2014.

**General comments**

As indicated above, Section 3 could be shortened considerable. In fact, I think the section would become much more clear if the final net topology and training are simply presented "as given". If there is any general experience to draw from the tests performed to reach the final configuration, summarise these separately. The present detailed description of the tests just obscures the final outcome, and it is very hard to extract if there is any experience of general interest.

*We are grateful to the reviewer for this good suggestion. We agree that the sections following 3.4.1 can be restructured and written more concisely in order to make the manuscript more clear and separate the training of CiPS from the comparison between different MLP structures. The Anonymous Reviewer #1 requested additional details about the determination of optimal meta-parameters, so in order to meet the requirements of both reviewers we have focused on restructuring and making this section more concise. First of all we have combined Sect. 3.4.2 and 3.4.3. We have also combined and shortened Sect. 3.5, 3.5.1, 3.5.2 and 3.5.3. We think that the comparison between different MLP structures will be interesting for readers that intend to develop ANNs for cloud remote sensing. However, those results are now presented separately at the very end of Sect. 3 ("Evaluation of different MLP structures") as the reviewer suggests. Please see the marked-up manuscript for details.*

[Figure]

*The preceding subsections in Sect. 3 (3.1–3.4.1) are already considered to be concise and we will keep them as they are unless the reviewer has specific suggestions on where we should shorten.*

Some comments on terminology used around the ANN training. In "machine learning" one is usually supposed to work with three datasets: training, validation and test set. The training set should be used to train one or several methods, validation set should be used to select the best one, and finally the test set should be used to evaluate the final system. In the manuscript the authors seem to mix up these sets. They use the validation set to monitor the training of the ANNs and then use the test set to select the best one and evaluate the performance. This is probably not critical for large datasets, nevertheless from a conceptual point of view this is not very nice. The authors refer to those datasets as training data, internal validation data, and (final) validation data, respectively.

Since this is also relevant for the follow-up paper, I suggest the authors create an additional test set that is used exclusively for evaluation.

*Again we thank the reviewer for raising an important point that was not described in a standard and clear manner in the manuscript. We do use the training, validation and test datasets that the reviewer refers to, and call them training, internal validation and final validation datasets in the manuscript. The reason for this was that in the meteorological field the term "test" has usually no clear meaning while the term "validation" has usually the meaning of the final evaluation of a dataset using an external independent dataset.*

*Furthermore, as the reviewer points out we falsely used the test dataset to select the optimal MLP structures after the first stage of the training as well as for the selection of the two classification thresholds. We have redone those steps using the validation*

*data and there are only marginal differences that don't affected our selection of MLP structures and classification thresholds. Thus, we modified the manuscript in this sense such that the test dataset is now used exclusively for the evaluation of the final ANNs. This makes the manuscript more correct from the conceptual point of view. Moreover, it is now clearly written that the validation datasets are used for the selection of MLP structures and classification thresholds and that the test dataset is exclusively used to evaluate the final performance.*

The authors should reflect upon if using CALIOP alone is the best choice for training data. Why not use some combined CloudSat and CALIOP retrievals, such as DARDAR? As far as I understand, the CALIOP lidar signal is quickly attenuated and I assume that the SEVIRI IR channels have a deeper penetration into the clouds. That is, CALIOP alone does not span a sufficient range of IWP. This problem should vanish if using e.g. DARDAR for training. This comment is a hint, no demand for redoing the work.

*Initially, we have had a synergistic CALIOP/CloudSat retrieval (e.g. DARDAR, 2C-ICE) as training reference data in mind. But using IR observations only, SEVIRI does not penetrate significantly deeper into the cloud than CALIOP does. Krebs et al. (2007), show that the radiative signal from the brightness temperature differences between the SEVIRI window channels (commonly used for cirrus remote sensing) peak at an optical thickness around 2–3 and is quickly attenuated for increasing values. Using a synergistic CALIOP/CloudSat retrieval, we would probably be able to retrieve larger optical thickness and IWP. However the point of saturation is not as clearly defined for a mixture of IR channels as for a laser beam. When CALIOP is the limiting factor we can easily identify those clouds where the optical thickness and IWP are untrustworthy (opaque clouds), which is the idea of our opacity classification ANN. If SEVIRI would be the limiting factor, it would be more difficult to identify the transparent clouds where both SEVIRI and CALIOP/CloudSat can retrieve trustworthy results and the retrievals*

[Figure]

*for thicker cirrus clouds would get ambiguous. Holl et al., (2014) use synergistic CALIOP/CloudSat retrievals to train an ANN, but to avoid too large retrieval errors for thicker clouds they also use additional measurements in the microwave range from MHS (microwave humidity sounder).*

*The following sentence has been added to the end of the concluding section: "Although this manuscript is limited to CALIOP retrievals, one could investigate the usefulness of synergistic CALIOP/CloudSat retrievals as training reference data."*

*Krebs, W., Mannstein, H., Bugliaro, L., and Mayer, B.: Technical note: A new day- and night-time Meteosat Second Generation Cirrus Detection Algorithm MeCiDA, Atmos. Chem. Phys., 7, 6145-6159, 2007.*

However, this touches upon the comment made above, that the errors of the CALIOP retrievals must be reported. These errors propagate directly into errors in the ANN product. As the test dataset is taken from the same CALIOP retrievals, the inherent CALIOP errors are not revealed. Conservative, quantitative, values on the dynamic range (i.e. coverage of optical thickness and IWP) and accuracy of the CALIOP product used for training shall be given.

*We agree with the reviewer that it should be clearly stated that the ANN will inherit all errors of the training data. We have revised the manuscript accordingly and this is now discussed in both Sect. 3.3 (Output data: cirrus properties from CALIOP) as well as in the concluding section. The accuracy of the CALIOP products was briefly summarised in Sect. 2.2 (CALIOP) in the initial manuscript. This part has been extended considerably and moved to Sect. 3.3 in order to make the link to the CiPS/ANN retrievals more clear. The CALIOP V3 optical thickness and IWP products have quality status "provisional", this means that only "limited comparisons with independent sources have been made and obvious artefacts fixed". A full error char-*

*acterisation of CALIOP is partly available in the literature and has been summarized in the revised manuscript (see below). Regarding the dynamic range of the CALIOP products, we consider it partly covered by Fig. 2, but additional details on the lower detection limit of CALIOP have been added in the revised manuscript. The following text segments have been added to Sect. 3.3 and to the concluding section respectively:*

[revised manuscript text omitted]

Further, there are also errors originating in the collocation procedure. Probably most important is the fact that CALIOP has a swath smaller than the resolution of SEVIRI. This results in that CALIOP covers only a part of the SEVIRI footprint, and this gives an additional uncertainty in the empirical relationships between CALIOP retrievals and SEVIRI measurements that the ANN is trained to represent. In any case, there are no comments at all of possible errors caused by imperfections in the collocation procedure.

*We thank the reviewer for this comment as this important point was indeed under-represented. We have added a paragraph where we discuss errors that should be expected as a result of the different spatial scales and observation techniques. The following text segment has been added to Sect. 3.4.1: "When collocating SEVIRI and*

*CALIOP observations with the purpose of training an ANN one must consider two aspects. 1) Even though the 5 km average of CALIOP point measurements fits the spatial resolution of SEVIRI ($3 \times 3$ km$^2$ at nadir and approx. $4 \times 5$ km$^2$ in mid-latitudes) quite well in the along-track direction, the two observations differ largely in scale in the across-track direction as the footprint of CALIOP is approx. 70 m wide at the Earth's surface. Consequently the 5 km CALIOP orbit segment is representative only for a relatively small fraction of a SEVIRI pixel. This will induce inevitable errors and lead to imperfect information used to train the ANN. This is especially relevant for partial cloud cover, where CALIOP may observe a cloud free area in an otherwise cloud covered SEVIRI pixel. If the error from imperfect collocations is random, this will have a limited effect on the ANN. Only if there is a recurrent systematic difference as a result of the different spatial scales this will be lead to biased retrievals (Holl et al., 2014). 2) Although cirrus clouds leave their mark on both SEVIRI and CALIOP measurements in a similar way, SEVIRI does not share CALIOP's possibility of discerning vertically separated ice clouds, liquid water clouds and aerosols. Consequently SEVIRI should not be expected to discern the signal from liquid water clouds and aerosols when retrieving the IOT as effectively as CALIOP."*

The points raised in the last two paragraphs, are they considered in the new manuscript targeting errors?

*No, they have instead been added to the revised version of this manuscript.*

*As ANNs do not provide a direct uncertainty estimate for the retrievals we have considered several approaches to characterise the errors/uncertainties, however we have not found an approach that is more representative than the overall statistics in this manuscript. Below we list some uncertainty estimate approaches together with their limitations. 1) One approach is to train a second set of artificial neural networks (ANNs) to retrieve the uncertainty reported by CALIOP, this would however require that*

[Figure]

*CiPS retrieves values that are identical or very close to those retrieved by CALIOP. As seen in the manuscript, this is not always the case. 2) A second approach is to run CiPS multiple times with small perturbations in the input data. This is however only representative if 100 % of the retrieval error can be attributed to noise and uncertainties in the input data. This is however not the case as most of the retrieval error of CiPS is likely to stem from the different sensitivities of SEVIRI and CALIOP. This is covered in the second manuscript, but then as a noise sensitivity analysis rather than an uncertainty estimate of CiPS. 3) A third approach is to train a second set of ANNs trained with the absolute difference between CiPS and CALIOP (using the same collocation dataset used to train CiPS). The retrieved accuracy would however be a statistical uncertainty learned using several training points and we consider the added value, in comparison to the characteristics on the overall performance presented here, to be small. So rather than estimating individual uncertainty measures of each retrieval we focus on performing a detailed characterisation of CiPS that we present here and in the second manuscript. Please note that most of this paragraph was taken directly from the reply to Anonymous Reviewer #1.*

**Specific comments**

p 5, l 23: Also the bias neurons need to be assigned the correct values.

*Revised.*

Sec 3.3: The section fails to clearly report what spatial resolution that is applied. For me this became clear first when reaching p 12, l 31. As 5 km anyhow is used, is the main discussion in Sec 3.3 actually needed? It seems to refer to an older version. That is, this section could be shortened.

*This was indeed not very clear. Since the Anonymous Reviewer #1 asked the same*

*questions we include the same reply here as well: We use the product with a reported spatial resolution of 5 km. But to detect faint cirrus and aerosol layers, the CALIOP team has to average over several consecutive 5 km profiles in order to get a sufficiently high signal-to-noise ratio. This means that in the 5 km cloud layer product, some cirrus were detected using a spatial resolution of 20 or even 80 km. In such a case the 5 km layer product will have 4 or 16 consecutive bins where the cirrus properties are identical. The additional spatial resolutions of 20 and 80 km can be seen as "background resolutions" used by the CALIOP team. This has been clarified by extending the paragraph, which now reads as follows: "Even though the cloud and aerosol layer product are reported with a spatial resolution of 5 km, two additional coarser resolutions of 20 and 80 km are used to detect the cloud and aerosol layers reported in the 5 km products (Vaughan et al., 2009). At a spatial resolution of 5 km, the signal-to-noise ratio of a faint cirrus or aerosol layer is usually too weak to be distinguished from the clear-sky atmospheric signal. By averaging 4 or 16 consecutive 5 km profiles the signal-to-noise ratio is increased, which allows for detection of very thin cirrus and aerosol layers. For example if a thin cirrus cloud with an optical thickness of 0.1 and a top altitude of 10 km is identified only when 16 consecutive 5 km profiles are averaged (80 km spatial resolution), 16 consecutive bins in the L2 5 km cloud layer data will report an optical thickness of 0.1 and a top altitude of 10 km."*

p 13, l 24: This is the only place where the authors mention the activation functions of the networks. This information should not be given below "Training data", but as indicated below. Further, why don't the authors use identity activation functions on the output layers of the regression ANNs instead of re-scaling, which would be the more common approach? This could also have an effect on the learning of extreme values since the gradient vanishes at both limits of the output range.

*The information about which activation functions are used has been removed from this section and shifted to the training section as proposed by the reviewer.*

*Our experience shows that using the identity function for output leads to competitive results. The only thing you gain is faster training in some cases, but training instability in others (especially if the range of the output data is not reduced). We do however avoid the extreme values when we scale the output data and use the intervals [0.1,0.9] for the sigmoid activation function and [-0.9,0.9] for the tanh activation function. We also tried more conservative scaling to the more linear part of the activation functions, but with no apparent improvement. This is equivalent to using a linear output as proposed by the reviewer. In our opinion the problem with the learning of extreme values would remain also with the identity activation function as this problem is caused by inevitable inconsistencies within the training data due to the different sensitivities of CALIOP and SEVIRI together with the fact that we minimise the squared error, meaning that the model can be wrong only in one direction at the extreme values.*

p 15, l 1-10: When introducing the structure of the networks the authors should mention which activation functions are used in the hidden layers and the output layer.

*Revised. The following sentence has been added to Sect. 3.5: "For the classification ANNs (CCF, OPF) the sigmoid activation function is used for both hidden and output layers, whereas the tanh activation function is used for hidden and output layers for the regression ANNs (CTH, IOT & IWP)."*

p 15, l 28-29: 1 CPU@3.4 GHz does not describe the computer sufficiently especially since ANN inference is highly parallelizable. The authors should at least give number of cores and processor model.

*Revised. This part now reads as follows: "using 1 core à 3.40 GHz, Intel Core i5-3570". We also added the following sentence to clarify that the computation time can be reduced: "ANN computations are highly parallelizable, meaning that the computation*

*time can be reduced significantly by distributing the computations across multiple cores."*

p 16, l 25: Figure 3 does not seem to compare the performance after the final training to the performance before, so the reference here seems pointless.

*After the changes made to Sect. 3 (see above), this reference no longer exists.*

p 17, l 5-13: The detection threshold for the CCF is a parameter of the classification ANN and is thus prone to overfitting. Its value should not be determined based on the performance on the test set. This touches upon the general comment above. Also,a plot of the POD against FAR for different thresholds would be good as it gives an additional perspective on the performance of the classifier.

*We thank the reviewer for pointing this out. As mentioned above we have redone this step using the validation dataset (referred to as internal validation data in the manuscript). The results are nearly identical to those we had with the test dataset (referred to as final validation data in the manuscript). Consequently, we choose the same thresholds as initially done with the test dataset i.e. 0.62 for the cirrus cloud detection and 0.86 for the opacity classification. The results are so similar that the numbers for the overall POD and FAR mentioned in this section remains the same.*

*A figure showing the POD and FAR as a function of the classification threshold (also know as the receiver operating characteristic (ROC) curve) has been added, as proposed by the reviewer.*

p 18, l 4: "the values corresponds" should be "the values correspond"

*Revised.*

p 18, l 9: Also here the authors claim the test data was excluded from the training but it seems that it has been used for the selection of the network structure and meta parameters.

*As explained above, the test dataset is now excluded from the training/development of CiPS and exclusively used for the final validation presented in Sect. 4.2.*

p 19, l 3 - 6: See comment p 17, l 5 - 13

*A reference has been added to the new figure showing the FAR and POD as a function of the classification threshold.*

p 21, l 5: "might seems high" should be "seem"

*Revised.*

p 21, l 7 - 14: The authors should explain in more detail what they mean with uncertainty and solution and/or provide a reference for their claims on the behaviour of ANNs.

*With **uncertainty** we mean the situation where a set of similar x-values, in the simplified ANN function f(x)=y, corresponds to quite different y-values. In such a situation it is not possible for the ANN to accurately model a relationship between x and y. Instead the ANN will output a mean value over the distribution of the most likely y-values that this x-value represented during the training, weighted by their probability. We call those y-values **solutions**. This segment has been rewritten and clarified and those terms are no longer used in the revised manuscript. Please see the response to p21, l 15 - 21 below.*

p21, l 15 - 21: Isn't that the reason for the "uncertainty" mentioned in the paragraph (see comment p21, l 7 - 14) in the training data? (I.e. low signal to noise ratio for thin clouds?)

*Yes, the reviewer is correct. This (the different sensitivities of CALIOP and SEVIRI) is indeed the reason behind the "uncertainty" mentioned above. We clarified this part accordingly and extended it with additional discussion and comments on the retrieval errors of CiPS, as suggested by the reviewer. Furthermore this part has been detached from the general discussion about the results and is now located at the very end of Sect. 4. Consequently, redundant parts, previously located earlier in that section, have been removed. The segment where the retrieval errors and the reason for systematic over- and underestimations are discussed now reads as follows: "As expected and as seen in Fig. 9, 11 and 12, CiPS is not able to perfectly model the CALIOP cirrus properties using the SEVIRI, ECMWF and auxiliary data. There are several sources of error that add to the final performance of CiPS. Most importantly CALIOP and SE-VIRI have different sensitivities to cirrus clouds. This is especially clear for thin to sub-visual cirrus clouds where CALIOP is able to accurately retrieve the top height and optical properties. Such faint cirrus leave a considerably weaker or no mark on the SEVIRI observations though, making it difficult to inversely determine the cirrus prop-erties. Similarly the CTH is not necessarily defined equally by CALIOP and SEVIRI, as CALIOP is able to discern thinner icy layers at the cloud top, that may appear as "invis-ible" to SEVIRI. Also for thicker cirrus clouds where both CALIOP and SEVIRI (thermal observations) approaches the point of saturation, the different sensitivities lead to am-biguous collocations. When an ANN is trained with a set of different output values that correspond to approximately the same input data as a result of the lower sensitivity, the ANN will not be able to model an accurate relationship. The reason for this is that the input vector contains no information on how the difference in sensitivity affects the tar-get values. This can be regarded as an unknown hidden variable. This is not an ANN*

*specific weakness, but applies to all regression models minimising the squared error. When such a set of incomplete input data (in the sense that there is a strong hidden variable) is given to the final ANN, it will output a conservative mean value that can be understood as an average over the distribution of the most likely solutions weighted by their probability. The larger the difference in sensitivity the higher will the variance within the distribution of the most likely solutions be, leading to larger retrieval errors. Throughout most of the output data range this error will be random. But obviously, the distribution of the most likely solutions cannot be centred around the extreme values leading to systematic over- and underestimations of low and high output values when a conservative mean value is calculated. This effect increases towards the extreme values as the desired output value is skewed towards the edge of the distribution of the most likely solutions. This effect is clearly seen in Fig. 11c and 12c where low and high $IOT_{CALIOP}$/$IWP_{CALIOP}$ are over- and underestimated respectively. This is to some extent also seen for the $CTH_{CiPS}$ retrieval in Fig. 9c, especially for low $CTH_{CALIOP}$. Due to the randomness of the effects a lower sensitivity introduces, adding information about the magnitude of the sensitivity to the input vector is not likely to improve this situation. The larger $CTH_{CiPS}$ retrieval errors observed for low clouds can also be attributed to the smaller temperature contrast with respect to the surface temperature and thus the weaker radiative signal that those clouds have compared to higher cirrus clouds. Another source of error that amplifies the effect discussed above, is the risk that there are additional variables relevant for finding an accurate relationship that are not represented by the vector of input data.*

*As discussed in Sect. 3.4.1 imperfect collocations as a result of the different spatial scales of CALIOP and SEVIRI together with partial cloud cover or spatially inhomogeneous clouds will further add to the retrieval errors. In a situation where CALIOP observed a small optically thin area of an otherwise optically thick cirrus inside a SEVIRI pixel, CiPS is likely to overestimate $IOT_{CALIOP}$ and $IWP_{CALIOP}$. Similarly if CALIOP observed a small optically thick area of an otherwise optically thin cirrus inside a SEVIRI pixel, CiPS is likely to underestimate $IOT_{CALIOP}$ and $IWP_{CALIOP}$."*